# High throughput screening identifies SOX2 as a super pioneer factor that inhibits DNA methylation maintenance at its binding sites

Ludovica Vanzan [1,4], Hadrien Soldati[1], Victor Ythier[1,5], Santosh Anand [1,6], Simon M. G. Braun [1], Nicole Francis [2] & Rabih Murr [1,3 ✉]

Binding of mammalian transcription factors (TFs) to regulatory regions is hindered by chromatin compaction and DNA methylation of their binding sites. Nevertheless, pioneer transcription factors (PFs), a distinct class of TFs, have the ability to access nucleosomal DNA, leading to nucleosome remodelling and enhanced chromatin accessibility. Whether PFs can bind to methylated sites and induce DNA demethylation is largely unknown. Using a highly parallelized approach to investigate PF ability to bind methylated DNA and induce DNA demethylation, we show that the interdependence between DNA methylation and TF binding is more complex than previously thought, even within a select group of TFs displaying pioneering activity; while some PFs do not affect the methylation status of their binding sites, we identified PFs that can protect DNA from methylation and others that can induce DNA demethylation at methylated binding sites. We call the latter super pioneer transcription factors (SPFs), as they are seemingly able to overcome several types of repressive epigenetic marks. Finally, while most SPFs induce TET-dependent active DNA demethylation, SOX2 binding leads to passive demethylation, an activity enhanced by the co-binding of OCT4. This finding suggests that SPFs could interfere with epigenetic memory during DNA replication.

---

[1] Department of Genetic Medicine and Development, University of Geneva Medical School, Geneva, Switzerland. [2] Institut de Recherches Cliniques de Montréal (IRCM) and Département de Biochimie et Médecine Moléculaire, Université de Montréal, Montréal, Canada. [3] Institute for Genetics and Genomics of Geneva (iGE3), University of Geneva, Geneva, Switzerland. [4] Present address: Institute of Bioengineering, School of Life Sciences, Ecole Polytechnique Fédérale de Lausanne (EPFL), Lausanne, Switzerland. [5] Present address: Diagnostic Department, Clinical Pathology Division, University Hospital of Geneva, Geneva, Switzerland. [6] Present address: Department of Informatics, Systems and Communications (DISCo), University of Milano-Bicocca, Milan, Italy. ✉email: rabih.murr@unige.ch

Transcription factor (TF) binding to specific sites at transcription regulatory regions is a fundamental step in gene expression regulation. However, as the sequence of a given regulatory region does not change between different cell types, the regulation of cell-specific transcription is likely to depend on non-genetic regulators. TF access to regulatory elements is controlled by the chromatin structure, which in turn is modulated by epigenetic modifications, such as nucleosome remodelling, histone modifications and DNA methylation. Epigenetic modifications are generally classified into active or repressive according to their effect on gene expression. DNA tightly wrapped around nucleosomes due to repressive histone marks was shown to be refractory to TF binding[1]. Therefore, the activity of nucleosome remodellers that increase chromatin accessibility was deemed necessary prior to TF binding to compact chromatin[2]. Several studies identified a special class of TFs, called pioneer transcription factors (PFs), that access their target sites in condensed chromatin. Such an event increases chromatin accessibility via nucleosome remodelling, thus facilitating the recruitment of settler TFs that are, otherwise, unable to access condensed chromatin[3–6]. These findings hint for a more complex relationship between epigenetic- and genetic-based mechanisms in transcription regulation than previously thought. It is therefore important to establish whether and when epigenetic mechanisms constitute a primary event in the regulation of transcription and when do they simply result from the genetic composition of the regulatory regions (i.e. the presence of TF binding sites (TFBSs)).

DNA methylation is an essential epigenetic modification that was originally hypothesized not only to inhibit the accessibility of DNA but also its affinity to TFs[7,8]. Recent studies have shown that not all TFs are sensitive to DNA methylation. Moreover, some TFs preferentially bind to methylated sites[9–11]. However, it is currently less clear whether those TFs that are able bind to methylated DNA can also lead to changes in the methylation status of their binding sites, and if so, how. More precisely, can PFs, in addition to their ability to remodel the nucleosomes, induce DNA demethylation?

Using a high-throughput explorative approach, our study methodically determined the ability of reported PFs to induce DNA demethylation at their binding sites in mouse embryonic stem cells (mESCs) and in vitro differentiated neuronal progenitors (NPs). Results show that, while many PFs do not affect the methylation status of their binding sites, a group of PFs that we call protective pioneer transcription factors (PPFs) prevent acquisition of DNA methylation, while another group called super pioneer transcription factors (SPFs) induce DNA demethylation at their methylated binding sites. Importantly, we show that, while most SPF-driven demethylation is Ten-Eleven Translocation (TET)-dependent, SOX2 (SRY (sex determining region Y)-box 2), an essential factor for the acquisition and maintenance of pluripotency[12], inhibits DNA methyltransferase 1 (DNMT1)-dependent maintenance of methylation during replication. This inhibition is amplified by the co-binding of OCT4. Finally, while PFs are important to enhance chromatin accessibility, our results indicate that this may not be achieved solely by PF binding and the resulting demethylation, as exemplified by ATAC-Seq experiments around single SOX2 and CTCF binding sites. Interactions with multiple TFs are likely to be important for generating accessible chromatin.

## Results

**Hi-TransMet: a high throughput assay for the analysis of TF binding effect on DNA methylation.** We developed a method called Hi-TransMet (high-throughput analysis of transcription factor effect on DNA methylation) to simultaneously assess the effect of several TFs on DNA methylation around their binding sites, at single base pair (bp) resolution, while reducing the effect of other TFs than the ones tested. Specifically, we used a transgenic mESC line, in which a targeting site composed of two inverted LoxP sites framing negative and positive selection genes (hygromycin and thymidine kinase), was engineered at the β-globin locus[13,14]. This locus is inactive in non-erythroid cells and was shown to not interfere with the methylation status of the targeting site. Using the recombinase-mediated cassette exchange (RMCE) approach[15], the targeting site can be replaced by any DNA fragment surrounded by two LoxP sites in a donor plasmid (Supplementary Fig. 1a). In particular, we opted to insert a bacterial DNA fragment, called FR1, in which we can include one binding site of a mammalian TF. Importantly, the fragment is unlikely to contain any mammalian TFBS due to evolutionary distance to mammals[13], thus inclusion of a TF binding motif within it allows us to assess whether the corresponding TF, in isolation from the effect of other TFs, can protect DNA from methylation (when FR1 is inserted unmethylated) or induce DNA demethylation (when FR1 is inserted upon in vitro methylation). Further advantage of using FR1 as the same genetic backbone for all tested TFs is the reduction of TF-independent background interference. FR1 (Supplementary Fig. 1b) has a CpG ratio of 3.6%, making it akin to a CpG island with intermediate CpG content[16]. Moreover, it was reported to get methylated when inserted in the RMCE site and to maintain its methylation when inserted after in vitro methylation (M.SssI methyltransferase treatment)[13].

**Validation of Hi-TransMet to study the effect of pioneer TFs on DNA methylation at their binding sites.** With the aim of identifying factors that can lead to protection from de novo methylation or to DNA demethylation, we focused on PFs, as their ability to access compact chromatin makes them ideal candidates for binding methylated DNA and affecting the methylation status of their binding sites. To address potential effect of PF binding on DNA methylation, we first analysed the methylation levels in ESCs of a set of genomic regions routinely identified as binding sites for several PFs, so-called mouse aggregate cistromes[17] (Supplementary Fig. 2). These results showed that common endogenous PF binding sites have lower methylation levels than the surrounding genomic regions, in line with previous studies[14,18–20]. While this analysis is informative, it remains correlative as it is unable to distinguish between two scenarios: (1) binding sites are demethylated prior to PF binding or (2) PF binding leads to DNA demethylation. To address these possibilities, we selected 27 PFs based on previously published reports of pioneering activity (Supplementary Table 1). Consensus wild-type (WT) binding motifs were extracted from public databases[21–23] or from studies that used chromatin immunoprecipitation–sequencing (ChIP-Seq) data for motif identification[24,25] (see "Methods"). Notably, these motifs were experimentally shown to be able to recruit their assigned PFs (Supplementary Table 2). One binding motif was selected for every PF with a few exceptions: GATA3, 4 and 6 share the same binding motif; SOX2 and OCT4 motifs were introduced either separately or in combination (OCT4SOX2) as the two factors were shown to often colocalize in ESCs[26–28] and during differentiation[29]; we also included two reported CTCF motifs (CTCF.1 and CTCF.2) corresponding to two different orientations within the genomic context, as the orientation of the CTCF motif was reported to affect its looping direction[30] (Supplementary Tables 1 and 2). For each WT motif, we designed a scrambled (Sc) control motif (Supplementary Table 2) with a significantly weaker binding score and in which only CpG

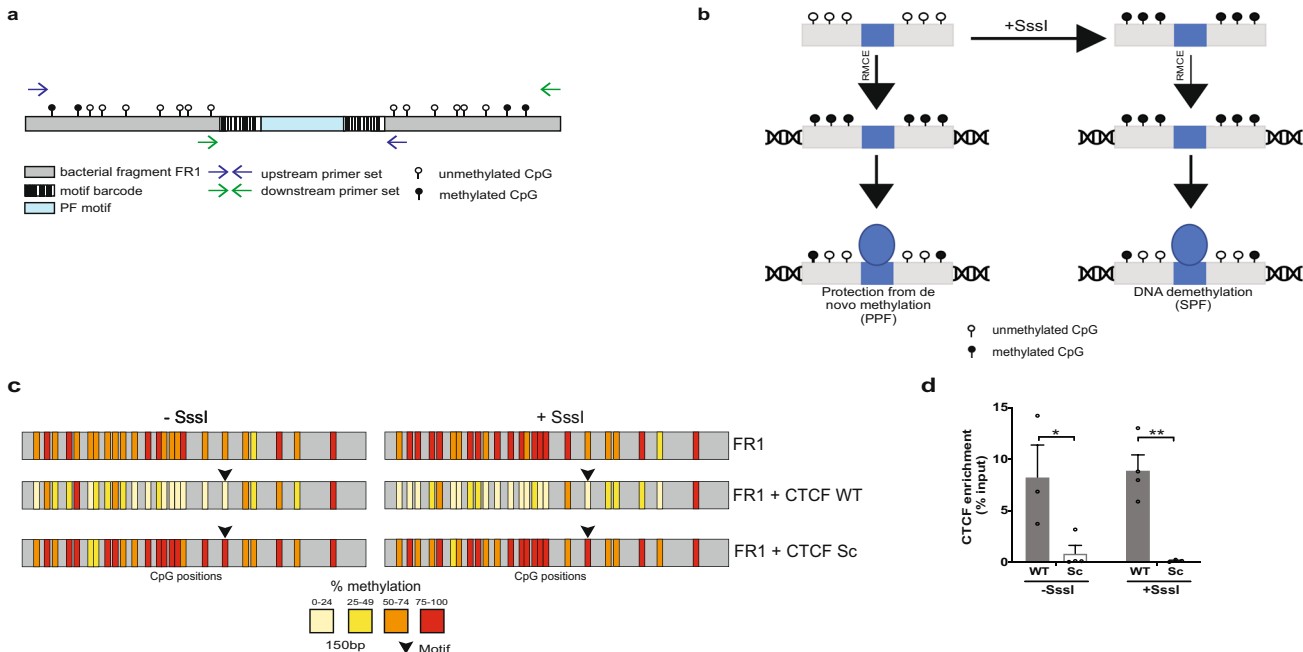

**Fig. 1 Validation of the experimental approach (Hi-TransMet) to test for TF effect on DNA methylation. a** PF motifs flanked by unique barcode sequences were individually cloned at the centre of an intermediate CpG content bacterial DNA fragment (FR1) within an RMCE donor plasmid that was used to insert FR1 in the target site of mESCs (**b**). Methylation levels in cells that underwent successful recombination were analysed using universal bisulfite PCR primers designed around the inserted motifs (blue and green arrows). **b** Schematic representation of Hi-TransMet screening principle. Motif-containing plasmid libraries were transfected into mESCs following in vitro methylation via M.SssI enzyme (+SssI) or without further treatment (−SssI). Upon insertion into the genome, untreated sequences gain methylation, while SssI-treated ones keep their methylation levels. Results allow to identify two types of PFs: (I) PFs that are able to bind unmethylated DNA and protect it from methylation (protective pioneer transcription factors (PPFs)); (II) PFs that are able to bind methylated DNA and induce DNA demethylation (super pioneer transcription factors (SPFs)). **c** Validation of the experimental approach by bisulfite Sanger sequencing in cell lines containing FR1 with CTCF-only motifs. Upon insertion, in the −SssI condition, FR1 undergoes de novo methylation. Lower methylation is observed in the presence of the CTCF WT but not Sc motif. In the +SssI condition, high levels of DNA methylation were retained by FR1 in the absence of the motif and in the presence of the CTCF Sc motif. In the presence of the CTCF WT motif, a reduction of DNA methylation levels is observed. Vertical bars correspond to CpG positions, and the colour code corresponds to the percentage of methylation calculated for each CpG with a minimum coverage of ten bisulfite reads. **d** CTCF binding to FR1 was verified by ChIP-qPCR. Results are presented as mean values + SEM of $n = 3$ (WT/ −SssI, Sc/+SssI) or $n = 4$ (Sc/−SssI, WT/+SssI) biologically independent replicates. Binding of CTCF to its motif in the FR1 fragment is indicated as percentage of input, that is, the enrichment of IP signal (normalized over input signal) at the selected locus. Enrichment at CTCF WT motif is significantly higher than at the Sc motif. $p$ values (two-tailed unpaired $t$ test): WT/−SssI vs Sc/−SssI:0.043494; WT/+SssI vs Sc/+ SssI: 0.004249 (\*$p < 0.05$, \*\*$p < 0.01$). Source data are provided as a Source data file.

positions present in their WT counterparts were maintained (see "Methods" for more details). Additionally, we assigned 6 bp barcodes on either side of each motif (Fig. 1a). One strand of these barcodes does not contain cytosines and is therefore not affected by bisulfite treatment, thus facilitating motif recognition following bisulfite conversion. We designed the barcode–motif–barcode combination by avoiding any resemblance to known TF motifs other than the ones intended (Supplementary Table 2 and "Methods"). In silico analysis of the selected barcode–motif–barcode combinations indicated that those containing WT motifs have little similarity to motifs of TFs other than the ones they are assigned to (Supplementary Fig. 3). On the other hand, we did not identify any significant known binding motifs in the combinations containing Sc motifs (Supplementary Fig. 3).

Double-stranded DNA oligomers, each representing a unique barcode–motif–barcode combination, were individually cloned into the FR1 within the RMCE donor plasmid (Supplementary Fig. 1b). Plasmids, each containing the FR1 backbone and one barcode–motif–barcode combination, were then mixed equimolarly to generate the targeting library that was further divided into two: one library was in vitro methylated (+SssI), while the second did not undergo any further treatment (−SssI). Libraries were separately transfected into the transgenic mESCs containing the

RMCE site, together with a plasmid expressing the CRE recombinase (Fig. 1a, b and Supplementary Fig. 1). This results in the replacement of the RMCE target site by the FR1 containing a unique barcode–motif–barcode combination, via homologous recombination between the LoxP sequences. As a library of plasmids was used, antibiotic selection resulted in a pool of cells each containing the FR1 with a different motif. Genomic DNA was then extracted from successfully recombinant cells and treated with sodium bisulfite, followed by amplification of FR1 using an approach that amplifies the site regardless of the identity of the inserted motif and labels each molecule with a unique molecular identifier (UMI)[31]. This allows quantifying the methylation of the original unamplified sequences exclusively, eliminating PCR biases (Supplementary Fig. 4a, b). Reduction of methylation at CpGs surrounding WT binding motifs, in comparison to those surrounding the corresponding Sc motifs, identifies PFs protecting DNA from methylation (−SssI condition) or inducing DNA demethylation (+SssI condition) (Fig. 1b).

To verify the functionality of Hi-TransMet and its ability to correctly identify factors whose binding could lead to lower methylation, we checked the methylation levels of the FR1 fragment including CTCF.1 WT and Sc motifs, as a similar experimental setting was previously used to study CTCF-

mediated DNA demethylation[14]. For this experiment, we used four transgenic ESC lines selected from the pool of transfected cells, each containing FR1 and one CTCF motif (WT −SssI, WT +SssI, Sc −SssI, or Sc +SssI) exclusively. Methylation analysis indicated that CTCF binding can both protect unmethylated sites from acquisition of de novo methylation (−SssI condition) and induce demethylation at methylated sites (+SssI condition) (Fig. 1c). Enrichment of CTCF binding at WT motifs was verified by ChIP (Fig. 1d). These results validate the use of Hi-TransMet to identify factors that protect DNA from methylation or induce DNA demethylation.

**Identification of pioneer TFs that can protect their binding sites from acquisition of DNA methylation**. To identify PFs that can protect their binding sites from acquisition of DNA methylation, we analysed reads generated by Hi-TransMet performed under −SssI condition in the pool of transfected ESCs. Methylation percentages were extracted for all CpGs within 300 bp upstream and 250 bp downstream of the motifs (Supplementary Fig. 4b). Methylation levels around WT and Sc motifs were first independently analysed (Fig. 2a). As previously reported[13], FR1 fragments not containing WT motifs become methylated upon insertion with an average CG methylation level of 52.4% around Sc motifs (Fig. 2a and Supplementary Fig. 5). While lower methylation levels are observed mostly around WT motifs, we observed some changes around Sc motifs too. To correct for these, we subtracted, for every motif, the methylation level of each CpG in the locus with the Sc motif from that of the same CpG in the locus with the corresponding WT motif ($\Delta$met = % met_WT − % met_Sc, Fig. 2b). We then classified the TFs based on their effect on DNA methylation using unsupervised hierarchical clustering, followed by the identification of statistically significant hypomethylated regions (HMRs) in WT conditions. HMRs are defined as regions of >50 bp and containing a minimum of 3 consecutive CpGs, each having a $\Delta$met of ≥10% (see "Methods" for more details).

Results show that PFs differ in their ability to protect DNA from methylation. Globally, in our experimental context, only few of the selected PFs show ability to protect from acquisition of DNA methylation. We called these PPFs. In addition to the previously reported CTCF[14] (CTCF.1 and CTCF.2) and NRF1[18], our results indicate that KLF4, KLF7, OCT4SOX2, SOX9, REST, OTX2, and E2F1 also protect against methylation (Fig. 2b and Supplementary Table 3). Moreover, the presence of SOX2 motif alone shows a tendency to protect against methylation, although this effect is amplified in the presence of an OCT4 motif (OCT4SOX2). It is important to note that all identified PPFs, with the exception of SOX9, were highly expressed in ESCs (Fig. 2c).

**Identification of SPFs that can induce DNA demethylation at their binding sites**. To identify PFs that can cause DNA demethylation upon binding to methylated DNA in ESCs, we analysed methylation levels in the +SssI condition. After insertion, the FR1 fragments not containing WT motifs maintained high levels of methylation in ESCs, with an average CG methylation level of 79.1% in fragments with Sc motifs (Fig. 3a and Supplementary Fig. 5a). Under these conditions, we observed extensive DNA demethylation around the WT binding sites of several factors: CTCF (CTCF.1 and CTCF.2), REST, KLF4, OCT4SOX2, SOX9, SOX17, E2F1, N-MYC, and GR (Fig. 3b and Supplementary Table 3). Moreover, a considerable, although not significant under our stringent cut-off, reduction of DNA methylation is again observed around the SOX2 motif, while this is less apparent around OCT4 motif. We called the corresponding factors SPFs as, in addition to their known ability to induce chromatin remodelling, they are also able to induce DNA demethylation. It is

interesting to note that CTCF, REST, SOX2, SOX9, E2F1 and KLF4 both protect from acquisition of DNA methylation and induce DNA demethylation. On the other hand, NRF1 and OTX2 can only protect DNA from methylation but have no effect on methylated DNA. This is in agreement with previously published studies defining NRF1 as methylation sensitive[18,32–34]. Similar to PPFs, most SPFs, with the exception of SOX9 and SOX17, are highly expressed in ESCs.

Interestingly, clustering of the results revealed that reduction in DNA methylation at some PPF and SPF binding sites extends far beyond the binding sites. This could be due to the sequence context of our reporter DNA fragment, which lacks motifs for other TFs, but might also suggest more active mechanisms, rather than steric hindrance, used by PFs to maintain low levels of DNA methylation and make a large region available for the binding of settler TFs.

To further assess the relative contribution of SOX2 and OCT4 in inducing DNA demethylation at their co-binding sites, we isolated, from the pool of transfected ESCs, a transgenic ESC line harbouring methylated FR1 (+SssI) and the OCT4SOX2 WT motif. We then used small interfering RNAs (siRNAs) to knockdown SOX2, OCT4 or both and performed methylation analysis around the OCT4SOX2 binding site contained in FR1. Globally, combinations of 2 siRNAs (siOct4 a + b and siSox2 a + b) were more efficient in reducing OCT4 and SOX2 expression than individual siRNAs (Supplementary Fig. 6). It also needs to be noted that, as the two proteins reciprocally regulate their expression[35], knockdown of one also affected the expression of the other. Overall, siSOX2 a + b siRNAs reduced SOX2 expression by ~45% and OCT4 expression by ~40%, while siOCT4 a + b siRNAs reduced OCT4 expression by ~80% and SOX2 expression by ~25% (Supplementary Fig. 6). Finally, combining all siRNAs reduced SOX2 expression by ~40% and OCT4 expression by ~75%. Methylation analysis revealed that all knockdown combinations increased the methylation level around the WT binding sites by ~2.5-fold relative to no-siRNA condition (from 16 to 38–40%—Fig. 3c), further confirming the direct role of SOX2 and OCT4 in inducing DNA demethylation at OCT4SOX2 binding sites. Accounting for the lower efficiency of SOX2 knockdown, the data suggest that, while both OCT4 and SOX2 contribute to DNA demethylation around their binding sites, SOX2 role is more prominent.

**Effect of PPFs and SPFs on DNA methylation correlates with their expression and activity**. To further confirm that the observed methylation changes are directly driven by the activity of the assigned PPFs and SPFs, we sought to measure the correlation between the expression level of these factors and their effect on the methylation status of their binding sites. We therefore induced differentiation of the transgenic ESC pool into NPs[36] (Supplementary Fig. 7c). Comparison of gene expression profiles derived by RNA sequencing (RNA-Seq) data in ESCs and NPs highlighted the differences in the expression of the tested PFs between the two cell types (Fig. 4a, b). Hi-TransMet was then performed in NPs and methylation levels around PF motifs in ESCs and NPs were compared.

First, differential expression of each tested PF between ESCs and NPs was plotted against the changes in $\Delta$met between ESCs and NPs of the FR1 containing the corresponding PF motif ($\Delta\Delta$met = $\Delta$met ESCs − $\Delta$met NPs). This showed an overall anticorrelation in both −SssI and +SssI conditions (Fig. 4c, d), indicating that methylation changes are indeed driven by the direct activity of the corresponding PFs.

We then sought to identify PPFs and SPFs in NPs. Average methylation levels highly increased during differentiation, reaching 81.7% around Sc motifs in the −SssI condition

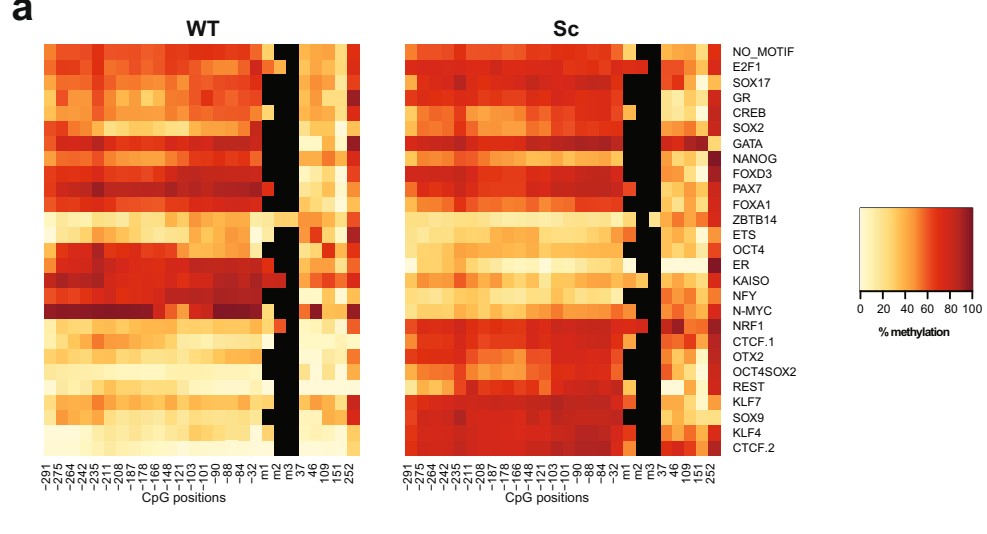

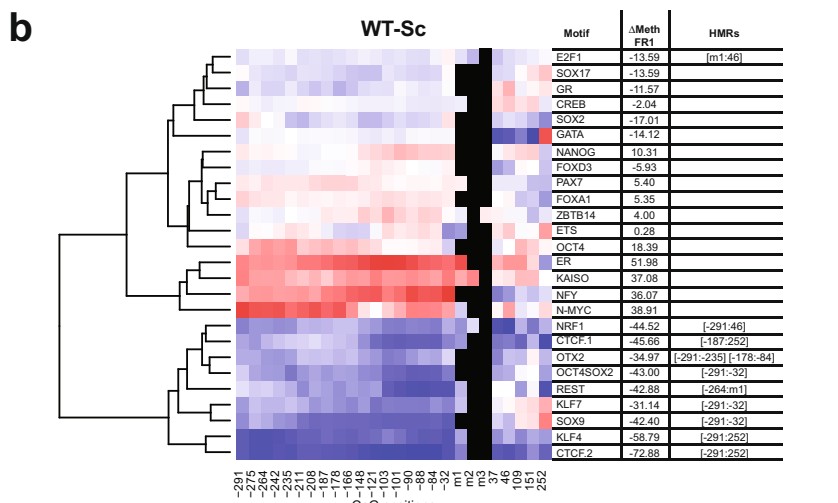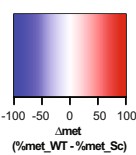

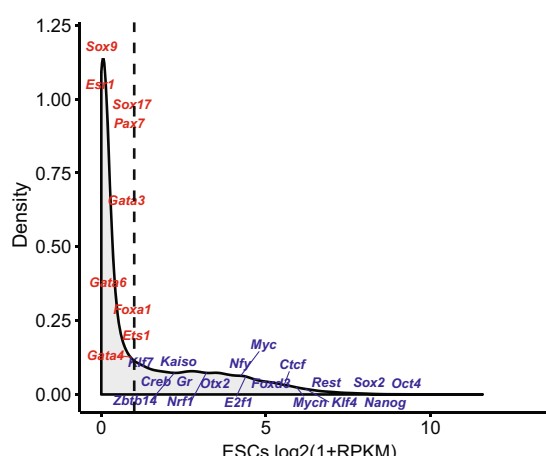

(Supplementary Figs. 5a and 7a) and 85.9% around Sc motifs in the +SssI condition (Supplementary Figs. 5a and 7b). In the −SssI condition, statistical analysis identified HMRs around CTCF.1, CTCF. 2, REST, KLF4, OCT4SOX2, SOX9 and N-MYC binding sites (Fig. 4e and Supplementary Table 3). On the other hand, FR1/+SssI data analysis identified CTCF, REST,

OCT4SOX2, SOX17, CREB, FOXA1 and FOXD3 as SPFs (Fig. 4f and Supplementary Table 3).

**SPF-dependent DNA demethylation is not sufficient to increase chromatin accessibility.** By definition, PF binding to

**Fig. 2 Identification of protective pioneer transcription factors (PPFs). a** Heatmaps indicating methylation percentages of individual CpGs in FR1 containing WT (left panel) and Sc (right panel) motifs in the −SssI condition. Each line represents FR1 containing the indicated motif. Each square within the line corresponds to one CpG. The methylation percentage of individual CpGs is represented by a colour code. CpGs' distance from the 5′ end of the motifs is indicated below the heatmaps. CpGs within the motif, when present, are indicated as m1, m2 and m3. **b** Differential methylation between WT and Sc motifs in the FR1/−SssI condition. Differential methylation was calculated for each CpG as Δmet = % met_WT − % met_Sc and represented by a colour code. Results were hierarchically clustered using the complete linkage method with Euclidian distance. CpGs' distance from the 5′ end of the motifs is indicated below the heatmaps. The coordinates of statistically significant hypomethylated regions (HMRs) in WT condition are indicated on the side. **c** Density plot of gene expression levels in mESCs as measured by RNA-Seq. Expression levels of the tested PFs are labeled. A cut-off of Log2(1 + RPKM) < 1 (dashed line) was used to separate PF expression levels into low (red) and high (blue).

condensed chromatin leads to increased accessibility of the surrounding region. Studies proposed several underlying mechanisms of this process: steric hindrance, recruitment of chromatin remodellers, or cooperativity with other TFs[37–43]. Our system has the advantage of allowing to assess the effect of PFs on chromatin accessibility, independent of their collaboration with other TFs, thanks to the absence of TFBSs in the vicinity of the tested motif. We therefore used ATAC-Seq to assess chromatin accessibility around OCT4SOX2 and CTCF binding sites within the FR1 (Supplementary Fig. 8). Results indicate that, while CTCF, OCT4 and SOX2 bind to their motifs, as evidenced by ChIP (Figs. 1d and 5c and Supplementary Fig. 9), and induce DNA demethylation, these events are not sufficient to generate chromatin accessibility of the surrounding region (Fig. 5a, b). While these results need to be reproduced on other PFs, they propose that collaboration with multiple sequence-specific DNA-binding factors is needed to result in changes in the chromatin structure, in agreement with previous studies[40–42].

**Most SPFs induce TET-dependent DNA demethylation**. Next, we sought to determine the mechanisms used by SPFs to demethylate their binding sites. DNA demethylation could occur in a replication-dependent fashion through the inhibition of the methylation maintenance machinery, notably the DNA methyltransferase DNMT1. Another possibility is that SPFs could induce replication-independent active demethylation processes.

TET-dependent oxidation of 5-methylcytosine (5mC) into 5-hydroxymethylcytosine (5hmC) is currently considered an essential step for active DNA demethylation. Several groups published interactions between PFs and TET enzymes[44–50] (Supplementary Table 1), consistent with studies reporting correlation between low levels of 5mC and high levels of 5hmC and TET proteins at TFBSs[44,51].

To address the functional involvement of TET proteins in SPF-dependent DNA demethylation, we performed Hi-TransMet on FR1/+SssI in mESCs lacking all TET proteins (TET1/2/3 triple knockout or TKO)[52]. In the absence of TET proteins, average methylation levels of FR1 are significantly higher than in ESCs expressing TETs, both in CG context (88.6% around Sc motifs, Fig. 6a and Supplementary Fig. 5a) and non-CG context (6.3% around Sc motifs, Supplementary Fig. 5b), suggesting that TET proteins are responsible for the majority of TF-independent demethylation events observed in the previous experiments. Moreover, most SPF-dependent DNA demethylation activity is weak or absent in TET TKO cells, indicating that SPFs mainly induce active DNA demethylation (Fig. 6b). Interestingly, demethylation still occurs in the absence of TETs at the OCT4SOX2 binding site. Although no statistically significant HMRs were identified, this is also observed at the SOX2-only binding site but not at the OCT4-only binding site. Other PF motifs that have lower methylation under these conditions are FOXD3, GATA and ETS1. GATA factors and ETS1 have very low expression in ESCs although they are slightly upregulated in TET TKO cells (Fig. 6c). It is therefore difficult to determine whether

the effect observed around their corresponding motifs is directly driven by these factors. On the other hand, FOXD3 is both highly expressed in TET TKO cells (Fig. 6c) and shows moderate SPF activity in NPs. SOX2 and FOXD3 might therefore lead to passive DNA demethylation, a possibility that we tested in the following paragraph. As TET TKO ESCs cannot be differentiated into NPs, NP-specific SPFs were also included in the following experiments aimed at testing SPF-dependent passive demethylation.

**SOX2 inhibits DNMT1 activity**. Maintenance of DNA methylation through cell replication is catalyzed by DNMT1. We therefore set up an in vitro methylation assay to assess the effect of PFs on DNMT1 activity[53]. A double-stranded hemi-methylated DNA probe containing the PF motif of interest and a single CpG (either within or in the immediate vicinity of the motif) was incubated with DNMT1 protein and radioactively labelled S-adenosyl-L-methionine (SAM[3H]) as a methyl donor, in the presence or absence of the corresponding PF. Integration of the radioactively labelled methyl group in the unmethylated strand was measured as a readout of DNMT1 activity, and for each PF, the signal in the presence of the WT motif was normalized to the signal in the presence of the Sc motif.

Results showed that only SOX2, and to a lesser extent OTX2 and ETS1, among all tested SPFs and non-SPFs, significantly reduce DNMT1 activity (Fig. 7a). Moreover, the presence of SOX2 alone, but not OCT4, is sufficient to significantly reduce DNMT1 activity on the DNA probe containing OCT4SOX2 motif, further confirming that SOX2 inhibits DNMT1 activity on hemi-methylated DNA (Fig. 7b). Unlike SOX2, FOXD3 does not affect DNMT1 activity. This suggests that FOXD3 might induce TET-independent active demethylation; however, further studies are needed to confirm this activity. Finally, NP-specific SPFs SOX17, CREB and FOXA1 do not affect DNMT1 activity, suggesting that they depend on TETs to induce demethylation.

**SOX2 inhibits DNMT1-mediated DNA methylation maintenance during replication**. To address whether SOX2-dependent inhibition of DNMT1 takes place during DNA replication, we set up an in vitro assay aimed at assessing the effect of TFs on the maintenance of DNA methylation during DNA replication[54,55] (Supplementary Fig. 10a, b). A bacterial DNA fragment containing the tested motif was cloned into an SV40 replication vector[56] to generate the replication substrate. Incubation of the substrate with T-Antigen, a replication cocktail and cellular extracts, allows its replication. Addition of biotinylated dUTP to the reaction marks nascent DNA with biotin, allowing streptavidin precipitation of replicated DNA. Complete replication was verified by digestion with the DpnI enzyme, which cuts specifically at GATC sites when the adenosine residue is methylated (m6A). As m6A is not maintained during replication, replicated templates are protected from digestion (Supplementary Fig. 10c). After immunoprecipitation with streptavidin beads, DNA methylation can be detected either by incorporation of SAM[3H] to the reaction or by bisulfite

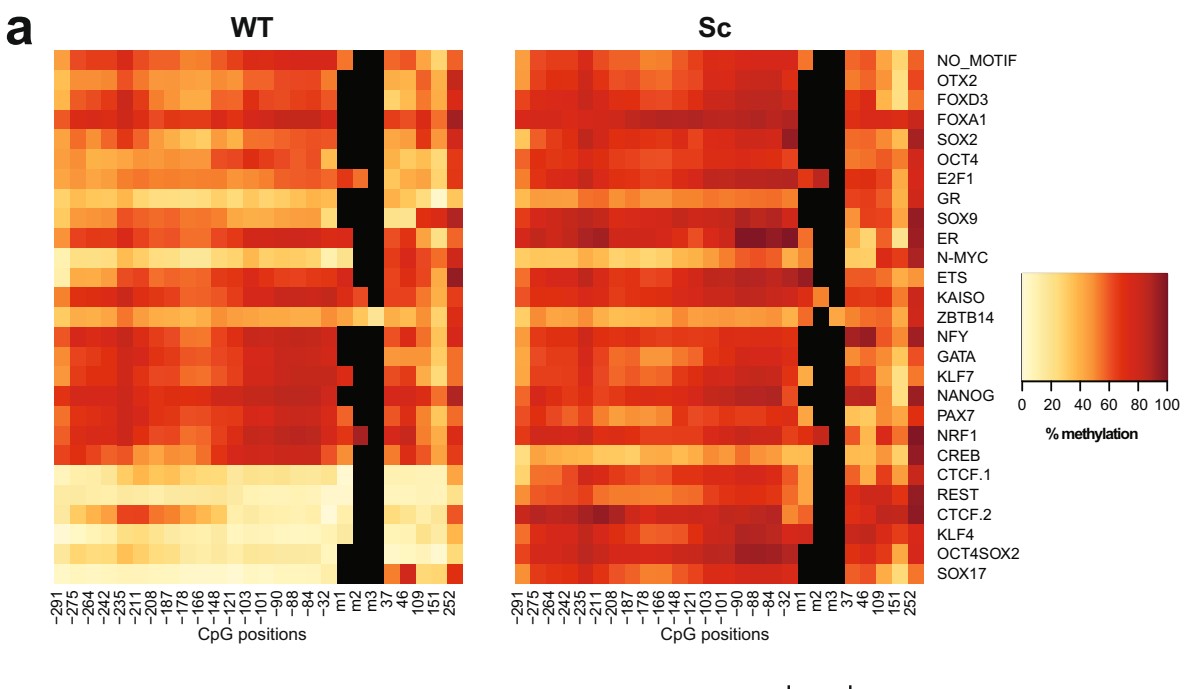

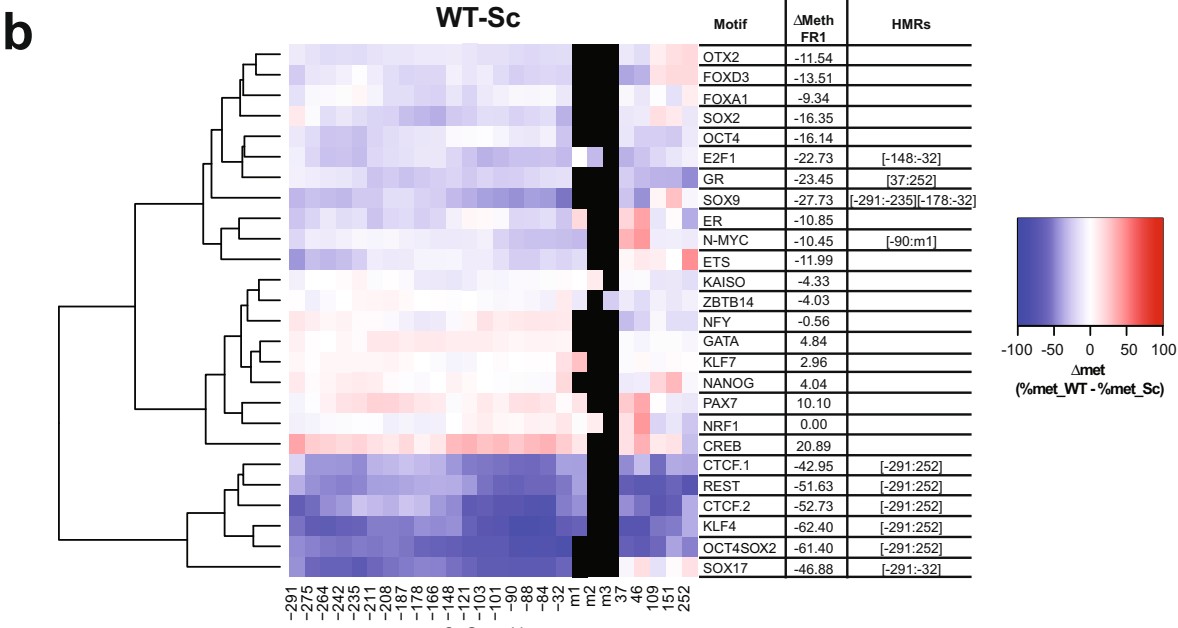

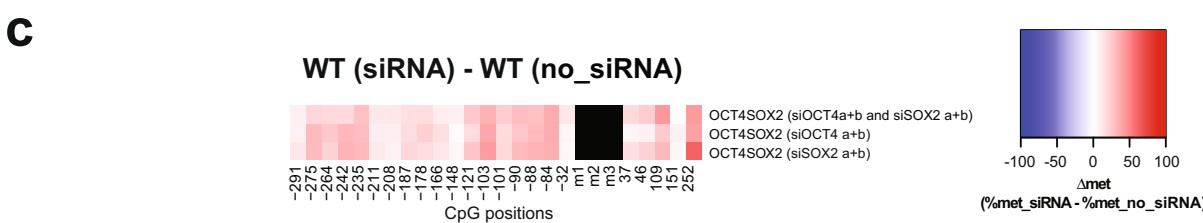

sequencing of the replication products. When the substrate is methylated by treatment with SssI prior to the replication reaction, maintenance of methylation is detected by incorporation of [3H]. In the absence of DNA methylation, no [3H] incorporation was detected, indicating that this system recapitulates maintenance of DNA methylation (Supplementary Figs. 10d and 11b).

We then performed the replication reaction in the presence or absence of the PF of interest. PF binding to the plasmid was verified by electrophoresis mobility shift assay (EMSA; Supplementary Fig. 10e); SAM[3H] incorporation was measured by scintillation counting and normalized to the signal in the absence of PFs. Results show that the substrate replicates efficiently both

**Fig. 3 Identification of super pioneer transcription factors (SPFs). a** Heatmaps indicating methylation percentages of individual CpGs in the FR1 containing WT (left panel) and Sc (right panel) motifs in the +SssI condition. Each line represents FR1 containing the indicated motif. Each square within the line corresponds to one CpG. The methylation percentage of individual CpGs is represented by a colour code. CpGs' distance from the 5′ end of the motifs is indicated below the heatmaps. CpGs within the motif, when present, are indicated as m1, m2 and m3. **b** Differential methylation between WT and Sc motifs in the FR1/+SssI condition. Differential methylation was calculated for each CpG as $\Delta$met = % met_WT − % met_Sc and represented by a colour code. Results were hierarchically clustered using the complete linkage method with Euclidian distance. CpGs' distance from the 5′ end of the motifs is indicated below the heatmaps. The coordinates of statistically significant hypomethylated regions (HMRs) in WT condition are indicated on the side. **c** Differential methylation around OCT4SOX2 WT motif in the FR1/+SssI condition between cells that underwent SOX2 and/or OCT4 knockdown (siRNA) and untransfected cells (no_siRNA). Differential methylation was calculated for each CpG as $\Delta$met = % met_WT (siRNA) − % met_WT (no_siRNA) and represented by a colour code. CpGs' distance from the 5′ end of the motifs is indicated below the heatmaps.

in the absence or presence of PFs (Supplementary Figs. 10c, d and 11c) and replicated templates maintain DNA methylation in the absence of PFs (Supplementary Fig. 10d). However, to rule out the possibility that differences in replication efficiency observed in the presence of PFs (Supplementary Fig. 11c) could affect the results, we normalized the methylation measures to replication efficiency (biotin-dUTP incorporation). In the presence of SOX2 protein, we observed a significant reduction in the methyl group incorporation both around SOX2-only and OCT4SOX2 WT motifs, indicating that SOX2 interferes with the maintenance of DNA methylation during replication (Fig. 7c and Supplementary Fig. 11a, b). Moreover, OCT4 does not inhibit DNA methylation maintenance during replication around the OCT4SOX2 motif (Fig. 7c), consistent with its inability to inhibit DNMT1. Results suggest a synergistic action of SOX2 and OCT4 in inhibiting DNA methylation maintenance, as shown after addition of both proteins to the replication reaction using the probe containing OCT4SOX2 motif (Fig. 7c). This is in agreement with the higher decrease of methylation observed by Hi-TransMet at OCT4SOX2 binding motif in comparison to SOX2-alone or OCT4-alone motifs. It is important to note that SOX2 activity does not seem to depend on the genomic context around its binding site. Indeed, experiments using FR2, another bacterial fragment with slightly higher CG content[13] than FR1, showed the same results (Fig. 7d). Analysis of the methylation status around the motifs by bisulfite Sanger sequencing of the replicated DNA displayed a reduction in DNA methylation in the presence of SOX2, but not of CTCF, FOXA1 and NFY, around their respective binding sites (Supplementary Fig. 11b), further confirming that SOX2 recruitment leads to passive DNA demethylation by DNMT1 inhibition.

## Discussion

While PFs' effect on nucleosome compaction is well documented, their crosstalk with DNA methylation is still poorly addressed. Here, we established Hi-TransMet, a high-throughput approach to assess the effect of TFs on DNA methylation. While we focused on PFs, this method could be used with any DNA-binding factor of interest and the throughput could be easily increased.

Using Hi-TransMet, we identified PPFs that are able to protect against de novo methylation. Our screening both confirms previously reported PPFs (NRF1[18,57], CTCF and REST[14]) and identifies new ones, either constitutive (KLF4, SOX2, SOX9) or ESC-(KLF7, E2F1 and OTX2) and NP-specific (N-MYC) (Supplementary Table 3). Whether PPF binding shields its surrounding from DNA methyltransferases by steric hindrance or whether PPFs directly interact with DNMT3a/3b/3L leading to their inhibition awaits further studies. We also identified SPFs that, in addition to their known pioneering activities, can induce DNA demethylation at their binding sites. Constitutive SPFs are CTCF, REST, SOX2 and SOX17. ESC-specific SPFs are KLF4, E2F1, GR, N-MYC and SOX9, while NP-specific SPFs are FOXA1, FOXD3 and CREB (Supplementary Table 3). It should be noted that the cell specificity of PPFs and SPFs is largely driven by changes in their expression level, thus

one could expect that these have the same effect in other cell types where they are expressed and active.

In agreement with our results, CTCF and REST were both predicted to induce DNA demethylation at their binding sites[14]. Similarly, several FOX factors were linked to DNA demethylation and TET1[58–61]. Moreover, overexpression of FOXA2, a paralogue of FOXA1, in fibroblasts correlates with chromatin opening and loss of methylation at its target sites. The pluripotency factor KLF4 was also recently shown to mediate active DNA demethylation at closed chromatin regions by interacting with TET2 during reprogramming[44]. Conversely, KLF7 was shown to interact with DNMT3a in TF protein array studies[62,63].

PPF and SPF activity may depend not only on their expression levels but also on their interactors, post-translational modifications and roles in different cell lines. For example, retinoic acid-mediated NP differentiation induces CREB phosphorylation, a necessary modification for its DNA binding ability, which could explain its NP-specific SPF activity[64,65]. Indeed, ChIP experiments show a preferential binding of CREB to its motif in NPs when compared to ESCs, despite its expression in both cell types (Supplementary Fig. 12a). While several studies have proposed CREB to be sensitive to DNA methylation[66,67], the lack of CREB phosphorylation in the assays used in those studies may explain this discrepancy. Also, although expressed at similar levels in ESCs and NPs, NRF1 and KLF7 lose their protecting ability in NPs. This might indicate a lower efficiency in stably protecting against methylation in conditions where higher levels of DNA methylation are the default status, as is the case in NPs. Similarly, N-MYC induces DNA demethylation in ESCs but only protects against methylation in NPs, suggesting that it is unable to demethylate highly methylated regions. Earlier reports on N-MYC are somewhat contradictory: on one hand, N-MYC was shown to be strongly linked to regions harbouring H3K4me3 histone marks, therefore most likely having low DNA methylation levels[68], and loss of N-MYC was associated with hetero-chromatinization in neuronal stem cells[69]. On the other hand, N-MYC was reported to bind to hypermethylated regions in neuroblastoma cell lines, although binding sites in this study lack the E-box CACGTG that was present in our study[70]. GR was previously shown to bind methylated cytosines in non-CG context; however, its effect on DNA methylation was not assessed[71]. Finally, SOX9 and SOX17 have low expression levels in ESCs but display a PPF and SPF behaviour in these cells. While low expression levels may be sufficient for this activity, it cannot be excluded that the motifs chosen might also be recognized and bound by other SOX family members, such as SOX3, 6 and 15 that have a very similar motif (Supplementary Fig. 3).

It is important to note that all WT motifs used here were previously tested experimentally for their ability to specifically and efficiently recruit their corresponding TFs, either by ChIP experiments or by DNA/protein microarrays (Supplementary Table 2). We therefore made the assumption that the TFs are able to bind to the WT motifs in our setting, and subsequently, lower DNA

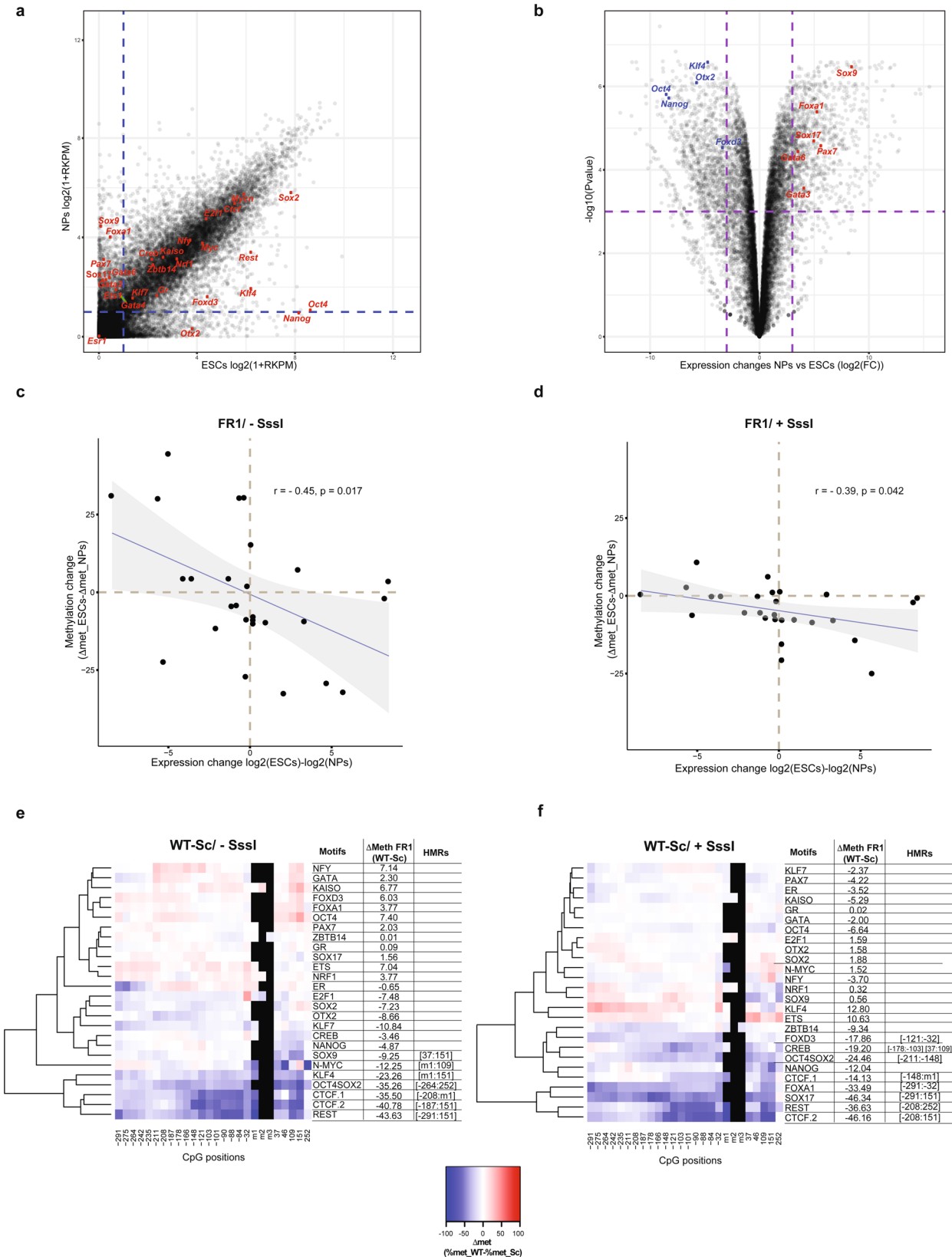

methylation levels around WT motifs in comparison to those around Sc motifs are attributed to this binding event. This assumption was validated by ChIP assays performed on selected PFs, PPFs and SPFs (Supplementary Fig. 12). Moreover, in the design step, we tried to minimize the possibility of unspecific bindings to the selected WT and Sc motifs. However, one cannot

completely rule out these events. Thus, Hi-TransMet is to be considered as exploratory and identified PPFs and SPFs should be confirmed in mechanistic assays as was done here with several of them.

While PFs are reported to enhance chromatin accessibility, our results suggest that the activity of a single SPF might be necessary

**Fig. 4 PPFs and SPFs are cell-type specific. a** Scatter plot showing differential gene expression between ESCs (*x* axis) and NPs (*y* axis) based on RNA-Seq data. Tested PFs are labelled in red. A cut-off of Log2(1 + RPKM) < 1 (dashed lines) was used to separate PF expression levels into low and high. **b** Volcano plot highlighting genes with a significantly different expression levels between ESCs and NPs. Cut-off is indicated by the dashed lines. **c**, **d** Scatter plots comparing differential expression of each tested PF in ESCs and NPs against the change in Δmet between ESCs and NPs (ΔΔmet = Δmet_ESCs − Δmet_NPs) of the FR1 containing the corresponding PF motif in −SssI (**c**) and +SssI (**d**) conditions. Each dot represents ΔΔmet for FR1 fragment with one motif. Data were analysed by two-sided Pearson's correlation test, with the error bands corresponding to the confidence interval. *r* = Pearson correlation coefficient; *p* = p value. **e**, **f** Differential methylation (Δmet) between WT and Sc motifs in the FR1/−SssI (**e**) and in the FR1/+SssI (**f**) conditions in NPs. Differential methylation was calculated for each CpG as Δmet = %met_WT − %met_Sc and represented by a colour code. Results were hierarchically clustered using the complete linkage method with Euclidian distance. CpGs' distance from the 5′ end of the motifs is indicated below the heatmaps. The coordinates of statistically significant hypomethylated regions (HMRs) in WT condition are indicated on the side.

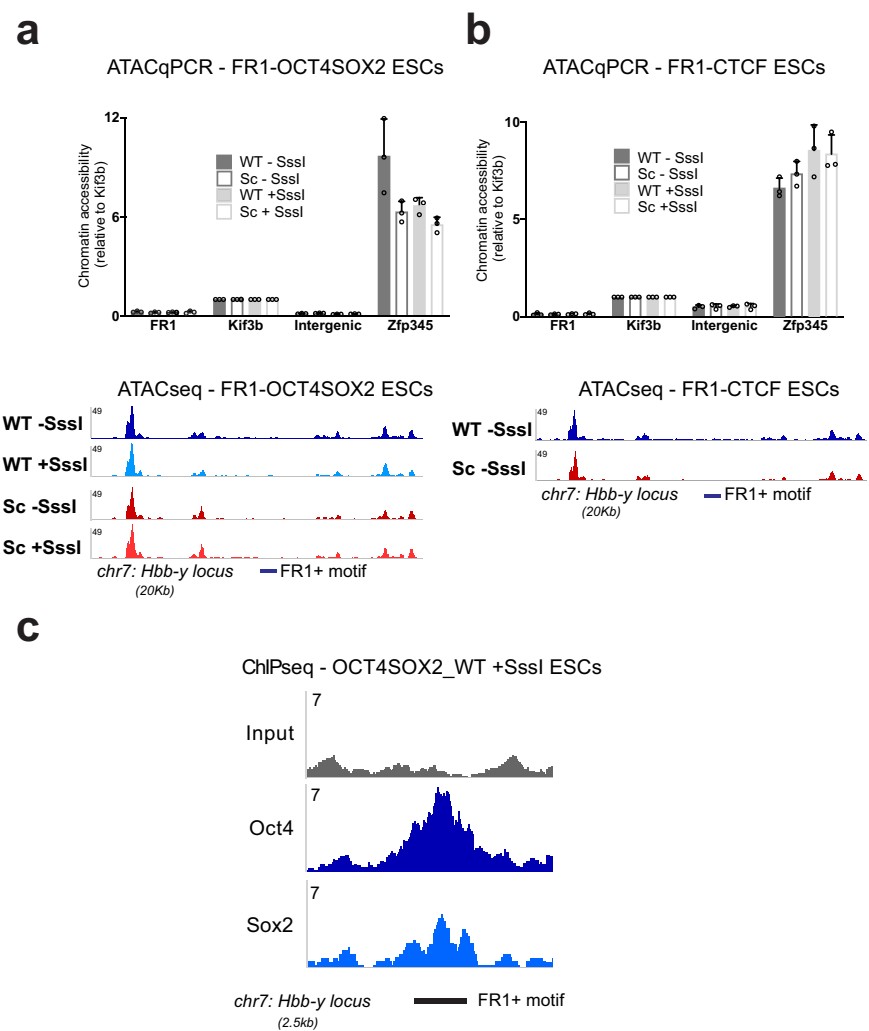

**Fig. 5 SPF activity is not sufficient, in isolation, to generate chromatin accessibility. a**, **b** Upper panels. Results of ATAC-qPCR experiments in FR1_OCT4SOX2 ESCs (**a**) and FR1_CTCF ESCs (**b**) containing WT and Sc motifs in −SssI and +SssI conditions. We measured very low levels of chromatin accessibility at the FR1 locus in all four conditions when compared to known accessible (*Zfp345* large ATAC-peak; *Kif3b* medium ATAC-peak) and inaccessible (*Intergenic* small ATAC peak) regions of the chromatin in mESCs[97]. Results are shown as mean+SD of *n* = 3 biologically independent replicates. Source data are provided as a Source data file. Lower panels. Representative genome browser tracks of the chromatin accessibility landscape around the FR1 locus. ATAC-Seq experiments were performed in FR1-OCT4SOX2 ESCs containing WT/±SssI and Sc/±SssI motifs (**a**) and in FR1-CTCF ESCs containing WT/−SssI and Sc/−SssI motifs (**b**). The ATAC-Seq signal is low over the FR1 locus compared to neighbouring accessibility peaks, highlighting the low levels of chromatin accessibility, which remain unchanged despite SPF binding in the WT conditions. **c** OCT4 and SOX2 ChIP-Seq in FR1 OCT4SOX2 ESCs reveals binding of both TFs at the FR1 locus. Displayed are representative genome browser tracks of the ChIP-Seq data across the FR1 locus.

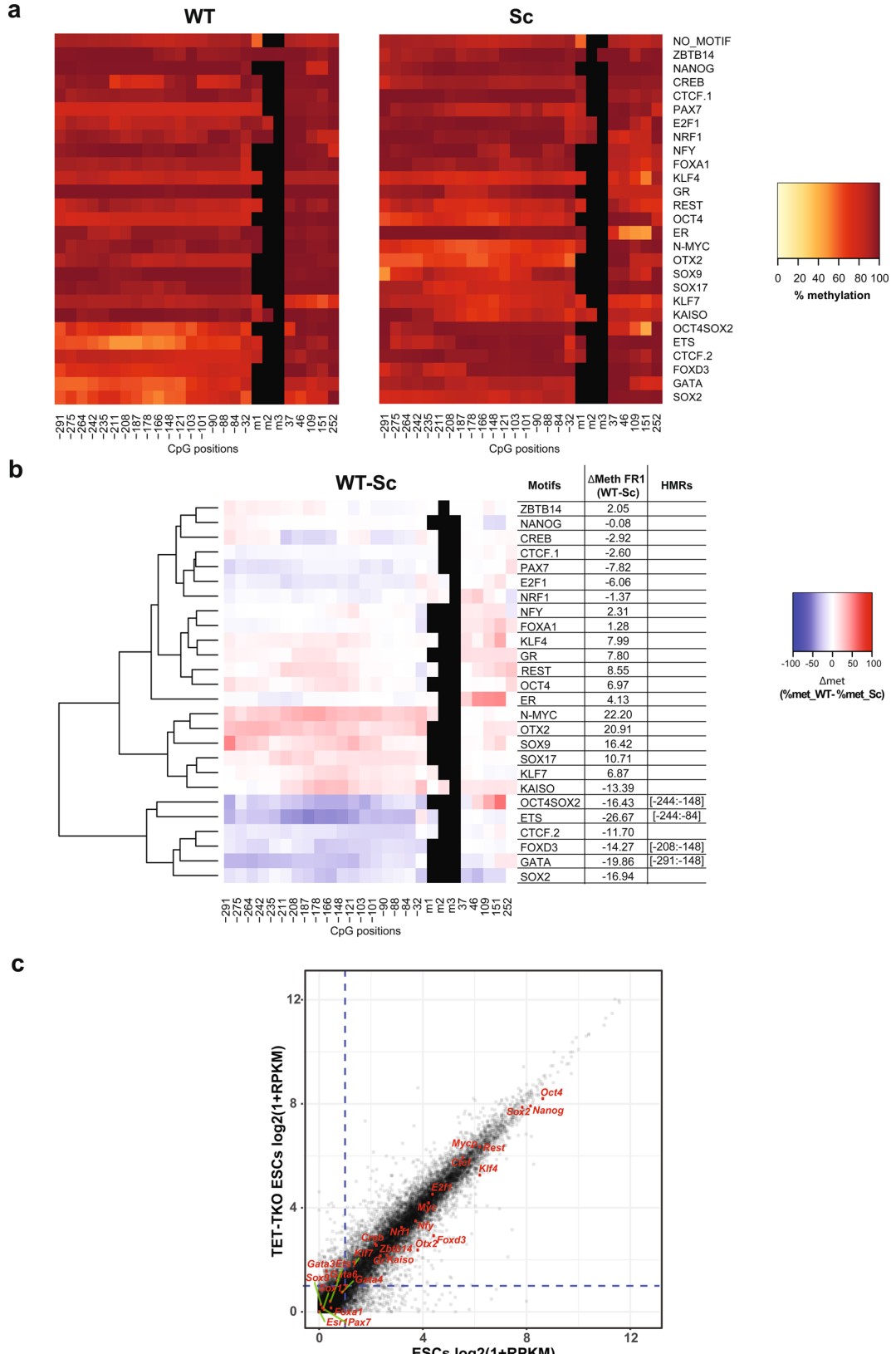

**Fig. 6 Most SPFs induce TET-dependent active DNA demethylation. a** Heatmaps indicating methylation percentages of individual CpGs in the FR1 containing WT (left panel) and Sc (right panel) motifs in the +SssI condition in TET TKO ESCs, as in Figs. 2a and 3a. **b** Differential methylation between WT and Sc motifs in the FR1/+SssI condition in TET TKO ESCs, as in previous figures. **c** Scatter plot showing differential gene expression between WT (*x* axis) and TET TKO (*y* axis) ESCs based on RNA-Seq data. Tested PFs are labelled in red. A cut-off of Log2(1 + RPKM) >1 (dashed lines) was used to separate PF expression levels into low and high.

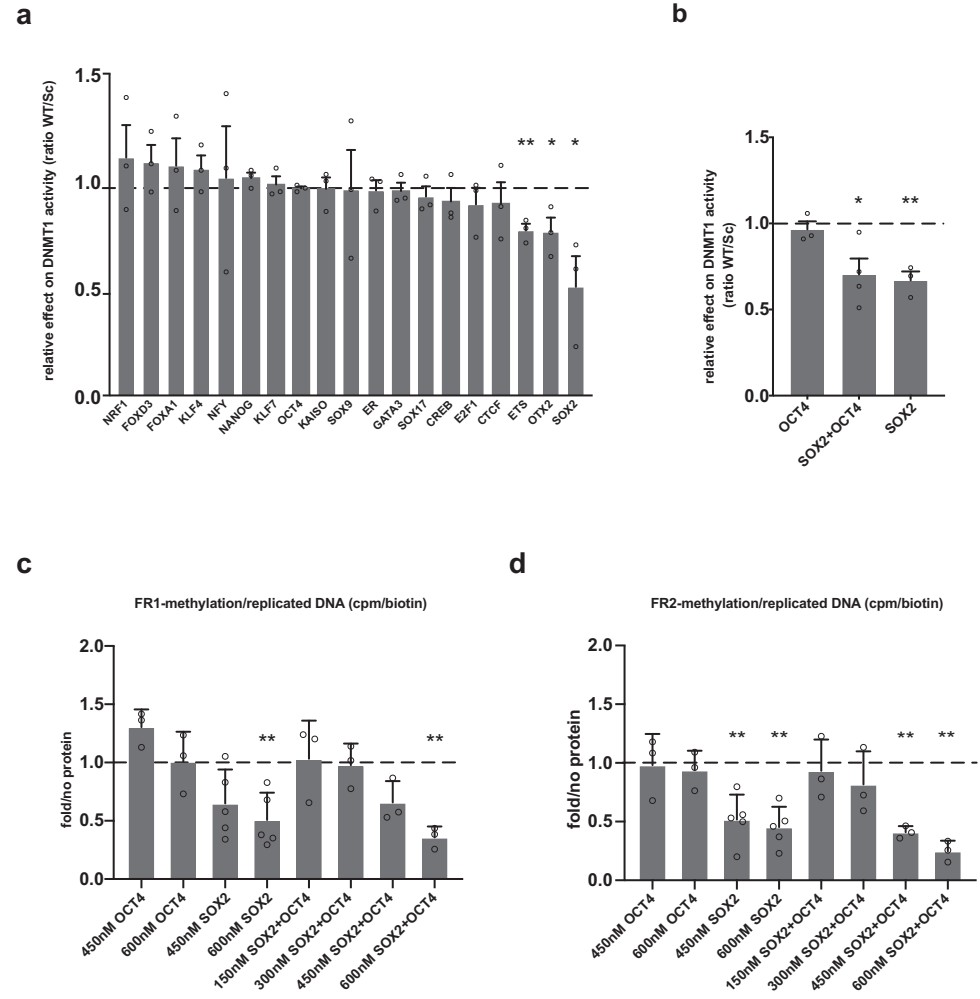

**Fig. 7 SOX2 inhibits DNMT1-dependent maintenance of DNA methylation during replication. a** In vitro methylation assay to measure DNMT1 methyltransferase activity using hemi-methylated probes containing WT or Sc PF motifs in the presence or absence of the corresponding PF. Relative DNMT1 activity is represented as scintillation counts in WT probes, corrected for the amount of recovered DNA probes and compared relative to the Sc probe. Results are shown as mean + SEM of $n = 3$ biologically independent replicates. SOX2, ETS1 and OTX2 significantly reduce DNMT1 activity. p values: ETS1 $p = 0.0018$, OTX2 $p = 0.0238$, SOX2 $p = 0.026$ (two-tailed unpaired t test; *$p < 0.05$, **$p < 0.01$). **b** In vitro methylation assay using the probe containing the OCT4SOX2 motif in the presence of OCT4, SOX2 or SOX2 + OCT4 proteins. Results are shown as mean + SEM of $n = 3$ biologically independent replicates. p values: SOX2 + OCT4 $p = 0.018$, SOX2 $p = 0.003$, (two-tailed unpaired t test, *$p < 0.05$, **$p < 0.01$). **c, d** In vitro replication assay to assess the effect of SOX2 and OCT4 binding on the maintenance of DNA methylation during replication. Two probes containing the OCT4SOX2 motif were used: FR1 (**c**) and FR2 (**d**). The indicated concentrations represent those of active hOCT4 and hSOX2 recombinant proteins that were used. Methylation levels following replication are measured based on the integration of radioactively labelled methyl group during replication and compared to "no_protein" control. Results are presented as mean + SD of $n = 5$ (hSOX2 samples) or $n = 3$ biologically independent replicates (hOCT4 and hOCT4 + hSOX2 samples) and analysed as radioactive signal in the presence of the protein relative to the signal in the absence of protein. p values: FR1_600nM_SOX2 $p = 0.0092$, FR1_600nM_OCT4 + SOX2 $p = 0.0068$, FR2_450nM_SOX2 $p = 0.0071$, FR2_600nM_SOX2 $p = 0.0021$, FR2_450nM_OCT4 + SOX2 $p = 0.0026$, FR2_600nM_OCT4 + SOX2 $p = 0.0047$ (two-tailed unpaired t test, **$p < 0.01$). For all panels, source data are provided as a Source data file.

but not sufficient to increase chromatin accessibility at its binding site. Indeed, ATAC-Seq experiments on OCT4, SOX2 and CTCF indicate that collaboration with other sequence-specific DNA-binding factors is likely required to open the chromatin (Fig. 5)[40,41].

Considering the previously established PF ability to access their binding sites in a closed chromatin context, the identification of SPFs introduces a further level of classification and suggests a hierarchy among TFs in the fine regulation of gene expression. We propose a model where SPFs are the first to engage methylated binding sites. This is followed by DNA demethylation allowing the recruitment of additional PFs that can bind to neighbouring sequences and allow for further chromatin opening, which provides access to settler TFs (Fig. 8).

OCT4 and SOX2 are known to widely colocalize in the genome of ESCs[29,72,73] and were reported to be involved in maintaining a hypomethylated state at the maternal *Igf2/H19* ICR, possibly through DNA demethylation[74,75] or by protection from de novo methylation[76], although a mechanism of action was not formally proposed. Our study shows a loss of methylation at SOX2 and OCT4 binding sites, although this was consistently more pronounced at SOX2 alone and further at OCT4SOX2 binding sites. It can be hypothesized that, while SOX2 is more efficient in mediating demethylation and protection from the acquisition of DNA methylation, OCT4's role is to cooperate with or to stabilize SOX2 binding, thus amplifying the effect on DNA methylation as observed at OCT4SOX2 binding sites. In NPs, OCT4 is silenced

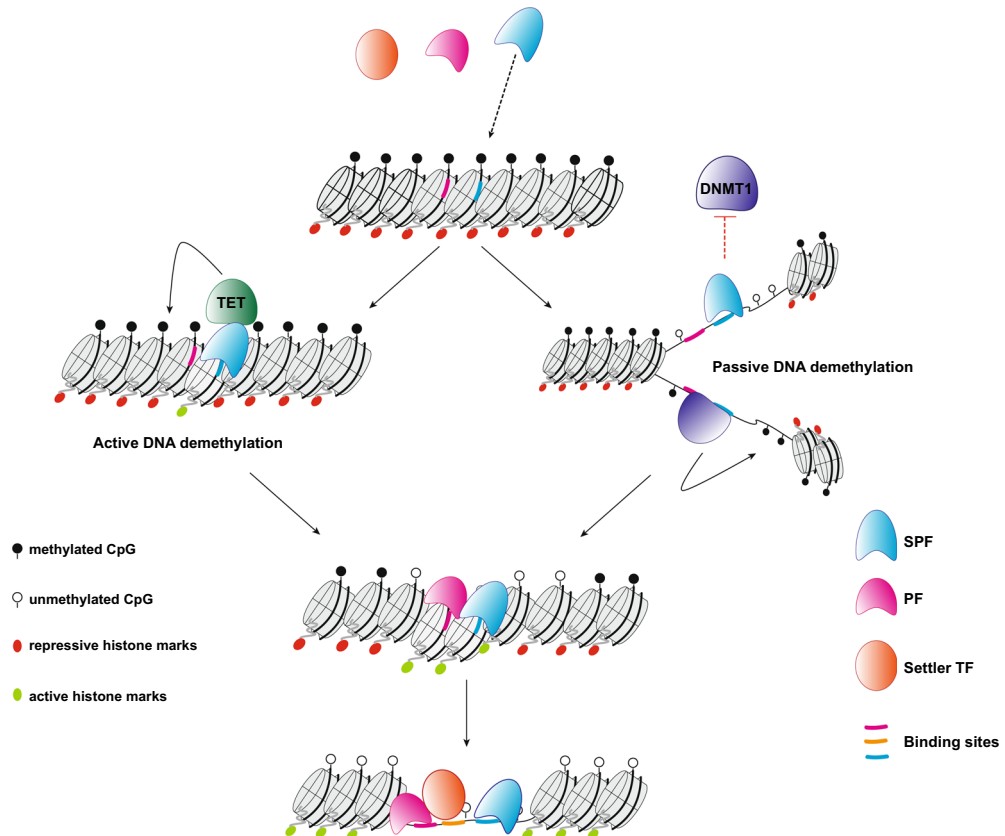

**Fig. 8 Hierarchy of transcription factor binding.** SPFs engage their target sequences in closed chromatin and in the presence of DNA methylation. Upon binding, most SPFs drive DNA demethylation through active processes, mainly mediated by the TET enzymes, while SOX2 leads to passive DNA demethylation. Loss of DNA methylation allows the binding of methylation-sensitive PFs. Nucleosome remodelling and deposition of histone modifications associated with open chromatin regions, mediated by both SPFs and PFs, create a favourable environment for the binding of settler TFs.

and replaced by the related POU family member BRN2 (POU3F2) in its interaction with SOX2[27]. Interestingly, the BRN2 binding motif is highly similar to that of OCT4, so it is plausible that the SOX2–BRN2 interaction in NPs has a similar effect on DNA methylation, which could explain the demethylation observed at OCT4SOX2 binding sites in NPs. This hypothesis awaits experimental validation.

We show that SOX2 mediates replication-dependent passive demethylation and that this activity is amplified by the presence of OCT4. Although replication was reported as necessary for TFs to induce DNA demethylation[77], our results provide evidence of direct TF interference with the activity of DNMT1 during replication in mammals. However, the exact mechanisms by which SOX2 mediates such an effect are yet to be elucidated. Based on the current knowledge, two possible mechanisms of SOX2-mediated passive DNA demethylation can be hypothesized: (1) SOX2 binding at the replication fork inhibits DNMT1 activity by steric hindrance. In this model, SOX2 binding would precede DNMT1 recruitment. This is conceivable as it was demonstrated that there is a delay in the recruitment of DNMT1 upon passage of the replication fork[78,79]. (2) SOX2 directly interacts with and inhibits components of the maintenance machinery. Indeed, a weak interaction between UHRF1 and SOX2 has been reported[80]. Finally, it would be interesting to assess the extent of this phenomenon and whether it is shared by other TFs. If it is the case, this could constitute another piece of the puzzle explaining the maintenance, or the lack thereof, of epigenetic modifications during replication.

## Methods

**Cell culture**. TC-1(WT) ES cells[13] were cultivated on dishes coated with 0.2% porcine skin gelatin (Sigma, cat. No. G1890) in high glucose–Dulbecco's modified Eagle's medium (DMEM; Gibco™, cat. No. 31966021) supplemented with 1% NAA (Gibco™, 11140035), 1:1000 homemade LIF and 1.42 nM beta mercaptoethanol. Differentiation into NPs was performed as follows[36]: $4 \times 10^6$ ESCs were grown in 15 mL CA (cellular aggregates) medium (DMEM, 10% foetal bovine serum (FBS), 1% NAA, 1.42 nM beta-mercaptoethanol) in non-adherent bacteriological Greiner Petri dishes (Greiner, Bio-one 94/16 with vents, 633102). CA medium was changed after 2 days of aggregation by transferring the CA suspension into a 50 mL Falcon tube, allowing CAs for 5 min. Medium is then aspirated and CAs are gently resuspended with 15 mL CA medium. Four days after plating, CA medium is changed and supplemented with 5 µM retinoic acid (Sigma, R-2625) for another 4 days. Medium was changed every 2 days. CAs were collected at day 8 after plating.

**Insertion of RMCE cassette in TET TKO ESC line**. Insertion of the RMCE Hy-TK cassette into the TET TKO mESCs was performed as follows[13]: $4 \times 10^6$ cells were transfected with 100 µg of the pZRMCE plasmid linearized with SapI (NEB, R0569S) using the Nucleofector™ 2b device and the Mouse ES Cell Nucleofector™ Kit (Lonza, VAPH-1001). The plasmid includes a 2.4- and 3.1-kb homologous arms to the positions −1300 upstream and +2332 downstream of the *Hbb-y* ATG start, respectively. These arms flank two inverted *LoxP* sites, which, in turn, flank the selection cassette. Upon transfection, positive selection of clones was performed using 25 µg/mL Hygromycin B Gold (InvivoGen, anti-hg-1) for 12 days. Surviving colonies were picked and screened for successful insertion by PCR (primer sequences in Supplementary Table 4).

**Recombinase-mediated cassette exchange**. Bacterial fragments FR1 or FR2 (corresponding, respectively, to FR9 and FR2 fragments in ref. [13]) were synthesized by Invitrogen GeneArt Gene Synthesis. FR1 was inserted into the RMCE donor plasmid by directional cloning using the restriction enzymes BamHI (NEB, R3136S) and HindIII (NEB, R3104L). Single-stranded oligomers containing the motifs were synthesized by ThermoFisher Scientific. For each motif, forward and

reverse oligomers were annealed and cloned into the FR1 fragment by directional cloning using the restriction enzymes SphI (NEB, R3133L) and NheI (NEB, R3131L). To create the plasmid libraries, single motifs containing plasmids were mixed in equimolar fashion and co-precipitated before RMCE or M.SssI treatment.

For RMCE transfection[81], ESCs containing the Hy-TK RMCE cassette were cultured in ES medium (15% FBS) containing 25 µg/mL hygromycin for at least 10 days and split the day before transfection. Medium was changed to 20% FBS ES medium 2 h before electroporation. Cells were then washed with phosphate-buffered saline (PBS), detached and counted. $12 \times 10^6$ cells were electroporated with 75 µg of the targeting plasmid or plasmid libraries and 45 µg of plc-CRE plasmid and plated in two P10 dishes with 20% FBS ES medium, as before. Positive selection with 3 µM ganciclovir (NEB, CLSYN001) was started 2 days after transfection.

For High-TransMet libraries, the pool of surviving cell was collected after 12 days, genomic DNA was extracted and efficient recombination was verified by PCR (primer sequences in Supplementary Table 4). For the generation of individual motif cell lines, after 12 days of selection, surviving colonies were picked and screened via PCR and Sanger sequencing in order to identify the inserted motifs.

**Plasmid methylation by M.SssI treatment**. When indicated, plasmid libraries were methylated before transfection using two rounds of treatment with the M.SssI CpG methyltransferase (NEB, M0226L)[82]. In each round, 100 µg of plasmids were incubated with 1× NEBuffer 2, 32 mM SAM and 22.5 µL (20,000 units/mL) M.SssI for 30 min. The reaction was then replenished with the same amounts of SAM and M.SssI in a final volume of 500 µL and incubated at 37 °C for another hour. Plasmid DNA was purified with phenol–chloroform and precipitated with ethanol. Complete methylation of the samples was verified by digestion with HpaII (NEB, R0171L), a methylation-sensitive restriction enzyme, and methylation-insensitive MspI (NEB, R0106L) as a control.

**Bisulfite conversion and PCR**. Genomic DNA was extracted using the GenElute Mammalian Genomic DNA Miniprep Kit (Sigma, G1N70-1KT). Bisulfite conversion of 800 ng of genomic DNA (for Sanger sequencing) or 3 µg (for Hi-TransMet library preparation) was conducted using the EZ DNA Methylation-Gold™ Kit (Zymo Research, D5006). Regions of interest were amplified by PCR using the AmpliTaq Gold™ DNA Polymerase (Applied Biosystems™, N8080241) and ran on a 1% agarose gel. Bisulfite PCR programme: 95 °C 15 min; 20 touch-down cycles from 61 to 51 °C with 30 s at 95 °C, 30 s annealing T and 1 min at 72 °C; 40 cycles of 30 s at 95 °C, 30 s at 53 °C and 1 min at 72 °C; final extension at 72 °C 15 min.

For Sanger sequencing, PCR products were extracted from 1% Agarose gels using the GenElute Gel Extraction Kit (Sigma, NA1111-1KT) and cloned into the pCR™4-TOPO plasmid of the TOPO® TA Cloning® Kit for Sequencing (Invitrogen, K45750), transformed into TOP10 bacteria, and plated on agar dishes with 100 µg/mL Ampicillin. Individual bacterial colonies were picked, followed by amplification and DNA extraction using the GenElute™ HP Plasmid Miniprep Kit (Sigma, NA0150-1KT). Finally, the products were sequenced using the M13r primer. Results were analysed using the BISMA [http://services.ibc.uni-stuttgart.de/BDPC/BISMA/] or BiQ [https://biq-analyzer.bioinf.mpi-inf.mpg.de/] Analyser online tools[83,84].

**Hi-TransMet library preparation and sequencing**. The UMI-based library protocol consists of 3 steps: annealing, non-barcoded amplification, and adaptors' addition (Supplementary Fig. 2a). For each library, 3 µg of bisulfite-converted DNA were used as starting material. Annealing programme: 95 °C 15 min; gradual temperature decrease from 61 °C to 51 °C, −0.5 °C/min; final extension at 72 °C 7 min. Annealing and extension were performed using the AmpliTaq Gold™ DNA Polymerase (Applied Biosystems™, N8080241), reaction set up according to the manufacturer's protocol. Following a purification step to remove unused primers, DNA was subjected to a short amplification with a universal forward primer and a specific reverse bisulfite primer: 95 °C 10 min; 3 cycles of 95 °C 15 s, 50 °C 30 s, 72 °C 1 min; final extension 72 °C 5 min. Amplified DNA was purified of the reaction mix, then sequencing adaptors were added in a final amplification step: 95 °C 15 min; 30 cycles of 95 °C 15 s and 60 °C 2 min. Primer dimers were eliminated in a final purification step. Primer sequences are listed in Supplementary Table 5. The PCR steps were done using the Promega Go-TaqG2 Hot Start Green Mastermix (Promega, M7423), set up according to manufacturer's protocol, and all the purification steps using the Qiagen GenRead Size Selection Kit (Qiagen, 180514). Correct library size was verified using the Agilent 2200 Tape Station system (Agilent, G2964AA, 5067–5584 and 5067–5585). Libraries were sequenced using Illumina MiSeq platform generating 300 bp paired-end reads (PE300).

**Chromatin immunoprecipitation**. ChIP was performed using the Diagenode IP-Star Compact Automated System robot (Diagenode, B03000002) and the Diagenode AutoiDeal ChIP-qPCR Kit standard protocol (Diagenode, C01010181) on $4 \times 10^6$ cells. Sonication was performed using the Diagenode Bioruptor Pico (Diagenode, B01060010) and the following conditions: 8 cycles of 30 s ON and 30 s OFF for mESCs; 10 cycles of 30 s ON and 30 s OFF for NPs. Correct DNA

fragments enrichment at around 200 bp was verified using the Agilent 2200 Tape Station system (Agilent, G2964AA, 5067–5584 and 5067–5585) and by gel electrophoresis. Three independent biological replicates were performed for each experiment; quantitative PCR (qPCR) was performed as described below. Antibody references and amounts used, according to the manufacturer's instructions: CTCF (Diagenode, C15410210, 1 µg), OCT4 (Diagenode, C15410305 and Cell Signaling Technology, 5677, 5 µg), SOX2 (Santa Cruz, sc-365823X, 5 µg), CREB (Abcam, ab31387, 5 µg), NRF1 (Abcam, ab55744, 8 µg), REST (Millipore, 17-641, 2 µg), NANOG (Cell Signaling Technology, 8822, 5 µL for $4 \times 10^6$ cells), NFY-A (Santa Cruz, sc-17753X, 5 µg). Primer sequences for qPCR are listed in Supplementary Table 4. ChIP-Seq libraries were prepared using the Illumina TruSeq ChIP Sample Preparation Kit (Illumina), and sequenced on an Illumina HiSeq 4000 sequencer.

**In vitro methylation assay**. Complementary single-stranded DNA (ssDNA) oligos, the forward strand containing one methylated CpG dinucleotide within or directly next to the PF motif, were synthetized by Microsynth AG. Hemi-methylated double-stranded DNA (dsDNA) probes were then produced by annealing these ssDNA oligomers. Annealed dsDNA probes were quantified using the Qubit™ 3.0 Fluorometer with the Qubit™ dsDNA HS Assay Kit (ThermoFisher Scientific, Q32854) and diluted to a final concentration of 800 nM. Reaction buffer was prepared as follows: 3 Ci/mmol SAM[3H] (Perkin Elmer, NET 155V250UC), 1× methylation buffer [40 mM Tris/HCl pH 7.5 (Invitrogen, 15504020), 10 mM EDTA (Applichem, A1104-0500), 10 mM dithiothreitol (DTT; Applichem, A1101.0005), 0.2% Glycerol (Sigma, 49767-1 L)], 0.2 mg/mL bovine serum albumin (Applichem, A1391.0500), 1× Protease Inhibitor cocktail (ROCHE, 05056489001). In all, 16.68 pmol of dsDNA probe were added to the buffer in three conditions: (1) buffer only, (2) buffer + DNMT1; (3) buffer + DNMT1 + 1× TF protein at an equimolar concentration to the probe. Samples were incubated at 37 °C for 1 h, then purified by phenol (Invitrogen 15513-039) and chloroform:IAA (Sigma, C0549-1PT) followed by ethanol precipitation. The DNA pellets were resuspended in 20 µL of TE buffer, then 15 µL of the eluate were placed on a filter paper and air dried; the remaining eluate was used to quantify the probe concentration for normalization. The filter papers were transferred into scintillation vials (Sigma, V6755-1000EA) with 4.5 mL of Ultima Gold Scintillation Liquid (Perkin Elmer, 6013151). Incorporation of $^3$H was measured on a Liquid Scintillation Counter (Wallac, 1409) for 5 min. The resulting measurements were normalized to the concentration of the eluate before further normalization to the baseline activity measured in the second condition containing only DNMT1. Recombinant proteins used in the assay were: DNMT1 (Abcam, ab198140), KAISO (Abcam, ab160762), ERα (Abcam, ab82606), NFYA (Abcam, ab131777), E2F1 (Abcam, ab82207), OCT4 (Abcam, ab169842), SOX2 (Abcam, ab169843), NRF1 (Abcam, ab132404), CTCF (Abcam, ab153114), FOXA1 (Abcam, ab98301), SOX9 (Abcam, ab131911), FOXD3 (Abcam, ab134848), KLF4 (Abcam, ab169841), ETS1 (Abcam, ab114322), KLF7 (Abcam, ab132999), NANOG (Abcam, ab134886), OTX2 (Abcam, ab200294), SOX17 (LSBio, LS-G69322-20), CREB (LSBio, LS-G28015-2), GATA3 (LSBio, LS-G67133-20).

**Protein production**. Recombinant proteins used in in vitro replication experiments were either purchased (Abcam: SOX2-ab169843 and OCT4-ab134876) or prepared in Sf9 cells. Baculoviruses for the expression of Flag-NFYA, Flag-FoxA1 or Flag-CTCF were used to infect 1 L of Sf9 cells for 48–72 h[85]. Cells were collected by centrifugation and washed with 5–10 volumes of PBS + 0.1 mM phenylmethane-sulfonylfluoride (PMSF). Cells were spun down, washed once with 1× PBS and pellets were resuspended in 2–3 volumes of Buffer F (20 mM Tris pH 8.0, 500 mM NaCl, 4 mM MgCl$_2$, 0.4 mM EDTA, 20% glycerol) plus NP40 to 0.05% with protease inhibitors (0.2 mM PMSF, 13.5 µM TLCK, 0.1 µM Benzamidine, 3 µM Pepstatin, 55 µM Phenanthroline, 1.5 µM Aprotinin and 23 µM Leupeptin), ZnCl$_2$ (10 µM final concentration) and DTT (1 mM final concentration). Cells were incubated on ice for 30 min and homogenized with a total of $3 \times 10$ strokes during the incubation. Extracts were centrifuged (30 min 48,000 × g), flash frozen and stored at −80 °C. For anti-FLAG affinity purification, extracts incubated with protease inhibitors and 1–2 mL of packed anti-FLAG resin (M2-agarose, Sigma), then binding was allowed to proceed overnight at 4 °C with rotation. Beads were centrifuged (1500 × g for 5 min), then washed with the following series: 2× Buffer FN, 2× BC1200N, 2× BC2000N (the second wash incubated for 15 min), 1× BC1200N, 1× BC600N, 1× BC300N, 1× BC300. The initial washes were carried out in batch (5 min rotation followed by centrifugation at 1000 × g for 4 min), and beads transferred to an Econo Column (Bio-Rad) at the BC2000N step. Proteins were eluted by incubation overnight with 0.4 mg/mL FLAG peptide in BC300 with 10 µM ZnCl$_2$ and protease inhibitors. Two additional elutions (with 1 h incubations) were collected. Eluted proteins were concentrated using Amicon Ultra 0.5 Centrifugal filter units (10 kDa molecular weight cut-off) (Millipore) and NP40 was added to 0.05% before aliquoting, flash freezing and storing at −80 °C. Protein concentration was determined by Bradford assay and adjusted for the purity as determined on SYPRO Ruby stained sodium dodecyl sulfate-polyacrylamide gel electrophoresis (SDS-PAGE) gels.

**In vitro DNA replication assay**. For large-scale DNA replication reactions used for bisulfite sequencing, TFs were pre-bound to 100–200 ng plasmid template containing FR1 or FR2 in 60 mM KCl, 12 mM Hepes pH 7.9, 2 mM MgCl$_2$, 1 mM DTT, 0.12 mM EDTA, 12% glycerol, 0.01% NP40 and 10 ng/µL DNA template for 15 min at 30 °C. For EMSA, 0.5 µL of each reaction (5 ng DNA) were removed, mixed with 4 µL of 50% glycerol/10 mM EDTA and loaded on a 0.8% agarose (SeaKem)/0.5× TBE gel, which was run for 90 min at 50 V. Replication mix was added to the remainder of the reaction. Replication mix consists of (per 100 ng DNA): 10 µL HeLa S240 extract, 1.38 µL replication cocktail (200 µM each rNTP, 100 µM dATP, dGTP, dCTP, 20µM dTTP, 40 mM phospho-creatine, 1 ng/µL creatine kinase (Sigma), 3 mM ATP, 5 mM MgCl$_2$), 0.2 µL human Topoisomerase II (TopoGen), 1 mM DTT, 0.32 µL Biotin-18-dUTP or Biotin-11-dUTP (1 mM, Jena Bioscience or Fisher), SAM[3H] (1 µCi/100 ng DNA, Perkin Elmer). Replication reactions were incubated for 90 min at 37 °C. Replication reactions were stopped with DSB-PK (5 µg/µL of proteinase K (Biobasic), 1% SDS, 50 mM Tris-HCl pH 8.0, 25% glycerol and 100 mM EDTA, digested overnight at 50 °C, followed by at least 30 min with RNaseA (1 µg/100 ng DNA) at 37 °C and purified by phenol–chloroform and chloroform extraction, followed by ethanol precipitation.

For binding to monovalent streptavidin beads (BcMag Monomeric Avidin Magnetic Beads, Bioclone Inc.), 40 µL of beads/750 ng reaction were prepared according to the manufacturer's instructions. Briefly, beads were washed 1× with 4 volumes of ddH$_2$0 and 1× with 4 volumes of PBS. All wash and binding steps were carried out at room temperature. Beads were incubated with 3 volumes of 5 mM Biotin (in TE-100), followed by washing with 6 volumes of 0.1 M Glycine pH 2.8. Beads were then washed twice with 4 volumes of TE-1000mM NaCl and added to purified DNA samples. One sample volume of TE-1000 was added to increase the [NaCl] to facilitate binding. Binding was carried out for at least 1 h and up to overnight with continuous rotation. Beads were washed three times with TE-100 and eluted 3 times with 75 µL mM Biotin in TE-100. Elutions were incubated at 50 °C with vortexing every 10–15 min.

To measure the incorporation of radioactive SAM[3H] during replication, reactions were carried out as above in the presence of SAM[3H] with 100 ng of template per reaction; all steps were scaled down linearly.

After digestion with proteinase K and RNaseA, DNA was purified using a PCR cleanup column, eluted in 45 µl and then passed through a G-50 spin column to remove any unincorporated SAM[3H]. SAM[3H] incorporation was measured by scintillation counting. DNA was quantified on agarose gels and used to normalize SAM[3H] incorporation. For the data shown in Fig. 7, we also quantified biotin-dUTP incorporation using a streptavidin slot blot to measure DNA replication and calculated SAM[3H]/biotin-dUTP. Both normalization methods produce the same conclusions.

**siRNA knockdowns**. siRNA knockdown experiments were conducted according to the manufacturer protocol for Silencer® Select Pre-Designed, Validated, and Custom Designed siRNAs (ThermoFisher Scientific). The following siRNA Silencer® Select oligos were used: Non-targeting (#4390843), siSOX2 a (s74175), siSOX2 b (s74176), siOCT4 a (s71992), and siOCT4 b (s17993). Briefly, ESCs were plated in 6-well plates at a density of 0.25–0.4 × 10$^6$/well. The next day, each well was transfected using Lipofectamine® RNAiMAX (ThermoFisher, 13778100) reagent with either mock (PBS) or 25 pM of non-targeting siRNA, siSOX2 a, siSOX2 b, siSOX2 a + b, siOCT4 a, siOCT4 b, siOCT4 a + b, siSOX2 a + b + siOCT4 a + b. Cells were collected 72 h post-transfection followed by DNA, RNA and protein extraction (see below). DNA was subjected to bisulfite conversion, Hi-TransMet library preparation and sequencing following the protocol mentioned above. RNA was used for cDNA preparation and quantitative reverse transcriptase–PCR (qRT-PCR) to assess SOX2, OCT4 and SNRPD3 (control) expression using primers in Supplementary Table 4.

**RNA extraction, cDNA preparation and RNA-Seq library preparation**. RNA was extracted using the Qiagen RNeasy Mini Kit (Qiagen, 74104) with the addition of the DNase step (RNase-free DNase set, Qiagen, 79254). RNA integrity was verified by running an aliquot on a 1% agarose gel. Conversion of 1 µg of RNA to cDNA was done using the Takara PrimeScript 1st strand cDNA Synthesis Kit (Takara, 6110A) according to the manufacturer's protocol. qRT-PCR was performed using the StepOnePlus qPCR by Applied Biosystems (ThermoFisher, 4376357) with the Applied Biosystems SYBR$^{TM}$ Green PCR Mastermix (ThermoFisher, 4309155). Primer sequences for qPCR are listed in Supplementary Table 4. RNA-Seq libraries were prepared from 500 ng of RNA using the TruSeq mRNA Stranded Kit (Illumina, RS-122-2101). Molarity and quality were assessed by Qubit and Tape Station. Biological replicates were barcoded and pooled at 2 nM and sequenced on 2 lanes using the Illumina HiSeq 4000 sequencer.

**Protein extraction and western blots**. Cells were washed twice with PBS, scraped with pre-heated Laemmli buffer (½ H$_2$O, ¼ SDS, ¼ Tris-HCl pH 6.8) and collected in an Eppendorf tube. Each lysate was boiled at 95 °C for 5 min and sonicated by passing the cells several times with through a 1-mL syringe. Protein concentrations was determined by BCA assay (Pierce, 23225). Proteins were separated by SDS-PAGE electrophoresis with a 4–12% Bis Tris gel (Thermo Scientific, NW0412C) and then transferred to a polyvinylidene difluoride membrane (BioRad, 162-0177).

Blots were probed with mouse anti-Oct4 (1:1000, Cell Signalling Technology, 5677), mouse anti-Sox2 (1:10,000, Santa Cruz, sc365823) and rabbit anti-Actin (1:1000, Abcam, ab8227) antibodies followed by horseradish peroxidase-conjugated secondary antibodies (1:10,000, BioRad, 170-6515 and 170-6516). Bands were detected following incubation with ECL (Pierce, 32106) by imaging on the iBright 1500 (Invitrogen) device.

**Motif design**. Criteria for choosing WT TF motifs were the following: (1) when available, motifs identified from ChIP-Seq data were selected. (2) if no such data are available, Position Frequency Matrices (PFMs) were obtained from the JASPAR Core Vertebrate 2016 database[21], alternatively, from the UniProbe[22] or TRANSFAC[23] databases. WT motifs were chosen mainly as the consensus sequence found in JASPAR database. To minimize the cross-matching between motifs, we checked that the WT core motifs (e.g. GAATGTTTGTTT) and the combination "restriction site–barcode–motif–barcode–restriction site" (e.g. catgtaGCATGCtgagaaGAATGTTTGTTTtgagaaGCTAGCcatgta) did not match with JASPAR motifs other than intended. This was done using the countPWM() function of the R BioStrings package using min.score = "90%". Scrambled (Sc) motifs were created by random shuffling of the WT motif except for the CG dinucleotides. The number and position of CG dinucleotides were maintained in WT and Sc motifs. For example, WT: **CCG**TAGTCGA and Sc: T**CG**AGCAGTC. Sc motif–barcode combinations were also checked for cross-matching with other JASPAR motifs as for WT sequences. To ascertain how closely the WT or Sc sequences match with the respective motif's PFM, a "normalized score" was defined. At each position in WT or Sc sequence, the probability of corresponding nucleotide in the PFM was taken as the match score for that position. The average of match scores for all positions was defined as "normalized score". Normalized scores of WT sequences were high (>0.7) and only those Sc motifs whose normalized score were at least 0.3 lower than the corresponding WT motif were used.

To confirm the specificity of the chosen motifs and that we are not creating any unwanted TFBS by adding the barcode and the restriction sites used for insertion, each motif–barcode–restriction site combination was screened using HOMER tools to predict for TFBSs. The non-redundant TF list from JASPAR 2018 database was used and log-odds score threshold was fixed to 6, above which a motif would be considered significant. Then TFBS predicted in WT sequences were normalized to those found in scramble sequences.

**Analysis of methylation status around PF aggregate cistromes**. Reads from ESC WGBS[14] were mapped on mm10 assembly using bitmapperBS[86] and methylation was extracted using MethylDackel [https://github.com/dpryan79/MethylDackel] tools. Aggregate cistromes (set of genomic regions routinely identified as binding sites) of the tested PFs were extracted from published data sets[17]. Only cistromes of category A (the highest reliability, experimental and technical reproducibility) and category B (high reliability, experimental reproducibility) were used. Bedtools[87] were used to get the methylation status 3 kb around the midpoint of every TFBS.

**Library data processing**. Paired-end libraries were trimmed using Trim Galore[88,89] and reads with a quality score <20 were discarded. Demultiplexing was performed with Flexbar[90,91], using the 6 bp library barcode plus 4 bp of the neighbouring adaptor for identification, with 0 mismatches allowed. Concomitantly, reads were tagged using the UMI-tags option of Flexbar based on the 8 UMI nucleotides that follow the library barcode sequence. Prior to mapping, motifs were extracted and PE reads were classified according to their motif using the vmatchPattern function with unfixed sequences (allowing IUPAC code for CpGs inside the motifs) from the BioStrings package designed for R[92], with 0 mismatches allowed. Reads were then mapped using Bowtie2[93] and Bismark[88], filtering out reads with non-CG methylation <2%. This filtering step was not performed for the libraries generated in TET TKO cells as the levels of non-CG methylation was significantly higher in these cells (Supplementary Fig. 5b). Reads were then deduplicated based on their UMI tag using the UMI_tools software[94] to remove PCR amplification biases. The percentage of methylation for each CpG position was extracted using Bismark, considering a minimum coverage of ten reads (Supplementary Fig. 4b). Biological and technical replicates were pooled to ensure sufficient coverage upon verification by Multi-Dimensional Scaling that replicates were clustering well together[95]. Ascending hierarchical clustering of the motifs, based on the methylation data, was obtained using the hclust function in R.

**ATAC-Seq and ATAC-qPCR**. ATAC-Seq libraries were prepared as follows[96]: 5 × 10$^5$ viable ESCs per sample were resuspended in 50 µL of transposition mix (Illumina) and incubated at 37 °C for 30 min. Libraries were amplified by PCR with barcoded Nextera primers and sequenced on an Illumina HiSeq 4000 sequencer. For ATAC-qPCR experiments, 1 ng of the ATAC-Seq libraries was used as input for the RT-qPCR reactions. FR1 and negative control primer sequencing are available in Supplementary Table 4; primers for Kif3b and Zfp345 control regions were taken from a previously published work[97]. ATAC-Seq reads were processed by mapping to the mm10 reference mouse genome (or to a 15 Mb region flanking the knocked-in FR1 sequence at the Hbb-γ locus) using Bowtie2[93]. Duplicate reads were discarded, leaving only unique reads. Peak calling was performed by MACS2.

Differential peak calls were made using DESeq2 and the requirement was set at a false discovery rate-corrected $p < 0.10$. RPM values in genome tracks are the mean values across replicates from each condition.

**RNA-Seq analysis**. SE 50 bp reads were trimmed[98] and then mapped to the mouse reference genome (GRm38.89 version from Ensembl) using the RNAseq aligner STAR[99] and featureCounts[100] to assign reads to their genomic features. Library size normalization and calculation of differential gene expression were performed using the edgeR package. Genes with a normalized maximal expression of >1 RPKM in all replicates were discarded. Fold change and Benjamini–Hochberg corrected $p$ value thresholds were set, respectively, to 3 and 1‰ for the differently expressed genes.

**Statistical analysis**. For the NGS data, methylation differences between WT and Sc for each CpG and for each motif were calculated using the DSS (dispersion shrinkage for sequencing data) package[101,102], with thresholds for Δmeth and corrected $p$ value fixed, respectively, at 10 and 5%. The percentage of methylation of each CpG was smoothed with adjacent CpG to improve mean estimation. The smoothing option was applied to a range of 50 bp. HMRs in WT vs Sc conditions (HMRs) were defined as regions of differential methylation of >50 bp and containing a minimum of 3 consecutive CpGs, each having a Δmet (%met_WT − %met_Sc) of ≥10%.

For in vitro methylation assays, three biological replicates were analysed using two-tailed unpaired $t$ test. In vitro replication results were analysed by one-sample unpaired $t$ test. qRT-PCR data for RNA expression in knockdown experiments were analysed using a two-way analysis of variance.

**Reporting summary**. Further information on research design is available in the Nature Research Reporting Summary linked to this article.

## Data availability

The data that support this study are available from the corresponding author upon reasonable request. All sequencing data generated in the course of this study (Hi-TransMet, RNA-Seq, ChIP-Seq, ATAC-Seq) were deposited in the GEO repository with accession number GSE144524. The mouse cistrome assembly and annotation data sets[17] are available in Figshare [https://doi.org/10.6084/m9.figshare.7087697]. Source data are provided with this paper.

## Code availability

Scripts are available at Github [https://github.com/MurrLabGEDEV].

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

## Acknowledgements

We thank M. Lorincz (University of British Columbia-Vancouver) for providing the RMCE plasmids, Ann Dean (National Institute for Diabetes and Digestive and Kidney Diseases) and D. Schübeler (Friedrich Miescher Institute) for the RMCE target ES cell line and R. Jaenisch (Whitehead Institute for Biomedical Research) for providing the TET-TKO ES cells. We are grateful to Sylvain Lemeille for help in designing the experimental approach and to Z. Herceg, D. Schübeler, T. Baubec and G. Andrey for helpful comments on the manuscript. Sequencing was performed at the iGE3 Genomic Platform of the University of Geneva. L.V. was supported by the iGE3. Research in the laboratory of R.M. is funded by the SNSF grants PP00P3_150712, PP00P3_179063, and PP00P3_190075; the Boninchi Foundation; Von Meissner Foundation and the Novartis Foundation for Medical-Biological research.

## Author contributions

R.M. and L.V. conceived the study; L.V., H.S., N.F., S.M.G.B. and R.M. performed experiments and analysed the results; H.S. and R.M. performed the in vitro methylation assays and contributed to the other experiments; N.F. performed the in vitro replication assays; V.Y., S.M.G.B. and S.A. performed the bioinformatic analysis; S.A., R.M. and L.V. designed the motifs; L.V. inserted the RMCE selection cassette into the TET-TKO ES cell lines and generated the individual motif cell lines; L.V., V.Y., S.M.G.B. and R.M. prepared the figures and wrote the manuscript with input from all authors.

## Competing interests

The authors declare no competing interests.
