## [Peer Review File · Nature Communications]

Reviewers' comments:

Reviewer #1 (Remarks to the Author):

This manuscript established a highly parallelized approach (Hi-TransMet) to investigate pioneer transcription factors' ability to bind to methylated DNA and induce demethylation. They found that a group of transcription factors can induce changes in DNA methylation at their binding sites. And they demonstrated that most pioneer transcription factors induce TET-dependent active DNA demethylation, while SOX2 binding leads to passive demethylation, which may suggest a novel mechanism allowing TFs to interfere with the epigenetic memory during DNA replication. The scientific question is important and the experimental design of this manuscript is relatively novel. However, several issues should be addressed.

Major points:

1. The authors need to offer more evidence to validate the reliability and effectiveness of the proposed method. The design of Hi-TransMet has a major defect that only one binding motif of every TF was selected without any evidence to confirm TF binding. The motifs selected were very likely to be bound by other TFs with similar motifs or off-targeted TFs, which led to DNA demethylation. Besides, the criterion to select the candidate motif seems not clear.
2. It's not clear why the authors identified SOX2 as super pioneer transcription factor but not OCT4 because only OCT4-SOX2 was reported in figure 2 and figure 3. The sentence in line 187, "Moreover, SOX2 alone, but not OCT4, seems to be able to protect against methylation, although this ability is increased in the presence of a combined OCT4-SOX2 motif", has no evidence to support.
3. In the fifth section, the author conducted differential expression analysis between ESCs and NPs and found PPFs and SPFs are cell-type specific. However, it seems that this chapter is not that closely related to subsequent chapters.

Minor points:

1. It's hard for readers to understand the method only by watching figure 1. Details or more labels may be appreciated.
2. In figure 1b, the authors should provide icon of unmethylated CpG and methylated CpG as well as Fig.1a.
3. The citation of figure 1c needs to be added to the corresponding main text.
4. Statistical test needs to be added to figure 1d.
5. In figure 2c,4a,4b and 5c, the gene name should be italic.
6. In figure 4, the words "Fig.4" are missed.
7. In figure 4g, the word "g" is missed.
8. In figure 5c and d, the figures are not aligned.
9. The X- and Y-axis should be marked in all figures.
10. In Supplementary table 1 and 3, in row 1, the content is in the middle of the table, however, in the rest rows, the content is left aligned.
11. In line 124, two different directionalities should be written as two different directions.
12. In line 186, "also" should be deleted.
13. In line 268, an extra punctuation mark is used.
14. In line 643, "WT: CCGTAGTCGA and SC: TCGAGCAGTC", are the bold letters necessary?
15. In line 653, "Library Data Processing" should be written as "Library data processing"
16. The logic discontinuous from "SOX2 inhibits DNMT1 activity" to "SOX2 inhibits DNA methylation maintenance during replication".

Reviewer #2 (Remarks to the Author):

Vanzan et al used a synthetic library approach to identify pioneer factors that lead to changes in the

local changes in DNA methylation. They carried out this experiment in ES cells as well as ES-derived NP cells. The authors then conducted genetic and biochemical experiments to probe the mechanism used by these factors to affect DNA methylation, in particular, their relation to TET and DNMT1.

There is an impressive amount of work in this study. The approach is novel, and the results are interesting. However, a few major concerns need to be addressed first:

1. The assumption that the PFs bind to their engineered motifs are concerning. It is known that TFs, and even PFs, only bind to a fraction of their recognition sites in the genome in a way that we do not fully understand. The ChIP study in Supplementary Figure 6 only tested factors that were tested positive in the assay. Even in these cases, significant binding of some factors (like NRF1) was not detected. The authors should test a few factors that did not show demethylation effect. If it turns out that these factors cannot bind to the engineered motifs, then the conclusion "most PFs do not induce changes in DNA methylation at their binding sites" are not valid.

2. Because the factor binding and motifs may not have one to one relation, knockdown experiment should be performed on selected factors. At least for Sox2, evidence should be presented that methylation level goes up after Sox2 knockdown.

3. There has been reports that the DNA methylation level is correlated with nucleosome occupancy. As pointed out in the manuscript, PF can lead to decreased nucleosome occupancy near its binding site. Could this effect a side effect of nucleosome occupancy change? If this question is beyond the scope of the current manuscript, at least some discussion should be made.

4. The effect presented are on the synthetic DNA. Does this effect apply to the native genome? Some bioinformatics analysis should be done to show if DNA methylation is protected at the binding sites of the identified factors.

5. The effect of Tet1 and DNMT1 are done with different method – genetic vs biochemical. Rationale should be provided why these two enzymes are treated differently.

6. For the biochemical reaction, how are the concentrations of the factor determined? Is there some estimate of the Kd?

Minor points:

There are many abbreviations in the manuscript that makes it hard to read. Terms like "triple knockout strain" or "hypomethylated regions", do they have to be shown in abbreviations?

Reviewers' comments:

Reviewer #1 (Remarks to the Author):

This manuscript established a highly parallelized approach (Hi-TransMet) to investigate pioneer transcription factors' ability to bind to methylated DNA and induce demethylation. They found that a group of transcription factors can induce changes in DNA methylation at their binding sites. And they demonstrated that most pioneer transcription factors induce TET-dependent active DNA demethylation, while SOX2 binding leads to passive demethylation, which may suggest a novel mechanism allowing TFs to interfere with the epigenetic memory during DNA replication. The scientific question is important and the experimental design of this manuscript is relatively novel. However, several issues should be addressed.

We thank the reviewer for appreciating the importance of the topic and the novelty of the design. Below are point-by-point answers to the reviewer's comments.

Major points:

1. The authors need to offer more evidence to validate the reliability and effectiveness of the proposed method. The design of Hi-TransMet has a major defect that only one binding motif of every TF was selected without any evidence to confirm TF binding. The motifs selected were very likely to be bound by other TFs with similar motifs or off-targeted TFs, which led to DNA demethylation. Besides, the criterion to select the candidate motif seems not clear.

This is a valid concern. As the reviewer suggested, it is difficult to completely rule out the possibility that the chosen motifs a) do not recruit the assigned TFs or b) recruit off-target TFs. We aimed at reducing these possibilities and their effects, both during the design step and by providing additional in silico and experimental evidence, as follows:

1) Experimental confirmation of the binding of TFs to their assigned motifs.

- We now performed more ChIP experiments on CTCF, CREB, FOXD3, E2F1, KLF4, NANOG, NFY, NRF1, OCT4, OTX2, REST and SOX2. While some of these experiments (E2F1, FOXD3, KLF4, OTX2) were unsuccessful due to lack of ChIP-grade antibodies, the rest (figures 1d, 5c, and supplementary figures 9 and 12) confirmed that SPFs specifically bind to methylated and unmethylated WT motifs (CTCF, SOX2, REST, CREB) while PPFs can bind to unmethylated WT motifs (CTCF, SOX2, REST). ChIP results also showed that TFs that do not belong to these two categories preferentially bind to their WT motif without affecting DNA methylation (e.g., Nanog and NFY). However, we do not rule out that certain PFs that are nor PPFs nor SPFs may not be able to bind to the inserted motif under our experimental conditions.*
- In addition to the ChIP assays, we investigated whether TF binding generates chromatin accessibility, with the aim of using accessibility as a proxy for TF binding and to explore potential chromatin remodeling activity. We therefore performed ATAC-Seq and ATAC-qPCR on RMCE cell lines containing CTCF or OCT4SOX2 motifs (WT/Sc/-Sssl/+Sssl). Results showed that although*

CTCF, SOX2, and OCT4 can bind their WT motifs (+/- Sssl, figures 1d, 5c and supplementary figure 12), they failed to significantly increase the accessibility of the chromatin at their binding sites. We conclude that this technique could not be used to detect the binding of the assigned TF to the inserted motif. More interestingly, these results indicate that 1) it is unlikely that additional TFs are binding around the inserted motif in an unspecific fashion, as this would likely have created an accessible site and 2) SPFs, in isolation, may not be sufficient to induce changes in chromatin accessibility and would need interactions with other TFs and/or chromatin remodelers, in agreement with previous studies (Chronis et al., 2017, Mayran et al., 2019, Sönmezer et al., 2020).

2) The choice of WT motifs: we selected motifs for which there exist the strongest experimental evidence supporting their ability to recruit the assigned TF. This was explained in the paragraph “motif selection” of the “online methods” section. We now added a supplementary table referencing this evidence (supplementary table 2).

In addition, we now assessed the likelihood that each sequence combination “restriction site-barcode-motif-barcode-restriction site” (restriction sites being those used for insertion of the motifs in the targeting plasmid) could contain sequences that resemble TFBSs other than the ones intended. Results of this analysis indicate that, while portions of these sequences may resemble other TF motifs, in the vast majority of cases the assigned TFBS is the highest ranking one (supplementary figure 3).

In few cases (CREB1, ETS1, KLF4, NMYC, OCT4, OTX2, and SOX2), other TFBSs also classified high. ChIP experiments performed on several of these TFs (CREB1, OCT4, and SOX2) confirmed the specific enrichment at WT motifs (supplementary figure 12). KLF4 and OTX2 ChIP, however, did not work, due to, respectively, the lack of good ChIP-grade antibody (two different ones were tested) and low expression in ESCs.

3) The choice of Sc motifs: in addition to the analysis summarized in the paragraph “motif selection” of the online methods section, supplementary figure 3 shows that all used sequence combinations “restriction site- barcode-Sc motif-barcode-restriction site” are unlikely to match to any known TFBS.

4) Finally, we would like to draw the reviewer’s attention to the fact that the FR1, in the absence of motifs (“no motif control”), shows higher methylation levels, indicating that the sequence around the inserted sites is unlikely to contain TFBSs that might result in the changes of methylation we observe.

2. It’s not clear why the authors identified SOX2 as super pioneer transcription factor but not OCT4 because only OCT4-SOX2 was reported in figure 2 and figure 3. The sentence in line 187, “Moreover, SOX2 alone, but not OCT4, seems to be able to protect against methylation, although this ability is increased in the presence of a combined OCT4-SOX2 motif”, has no evidence to support

The reviewer is right in pointing out that the distinction between the activity of SOX2 and OCT4 is not evident. Indeed, we could not identify, under our stringent cutoff, significant HMRs around the binding sites of both OCT4 and SOX2 (Figures 2 and 3). However, in figures 2 and 3, we observed lower methylation levels around the SOX2 binding site, when compared to the methylation levels around OCT4 binding site. This is what we reported in the sentence in question.

Furthermore, we still observe demethylation around the OCT4SOX2 motif in NPs (figure 4), although OCT4 expression is very low in NPs. This again points to a more prominent role of SOX2 in protecting from de novo methylation (PPF) or inducing DNA demethylation (SPF).

To further specify the roles of OCT4 and SOX2 in this process, we knocked down OCT4, SOX2 or both in the RMCE cell line containing the methylated FR1 and OCT4SOX2 motif, in +Sssl condition (supplementary figure 6), followed by Hi-TransMet (figure 3c). Data showed that reduction of expression of OCT4, SOX2 or both leads to increased methylation levels at the FR1 containing the OCT4SOX2 binding site. These results indicate, as rightly noted by the reviewer, that both OCT4 and SOX2 have a role in inducing DNA demethylation at their common binding sites. However, as SOX2 knockdown was less efficient than OCT4 knockdown, but the re-methylation levels were similar in both cases, these results still point towards a more prominent role for SOX2 in inducing DNA demethylation at OCT4SOX2 binding sites.

We also performed in vitro replication assays on probes containing the OCT4SOX2 binding motif embedded into two different bacterial backbones (FR1 that was used in the Hi-TransMet and FR2, a bacterial DNA fragment of slightly higher CG content) in the presence of OCT4 protein, SOX2 protein or both (figures 7c and 7d). Results clearly indicated that SOX2 is sufficient to efficiently inhibit the maintenance of DNA methylation during replication around OCT4SOX2 binding sites, while OCT4 is not. The presence of both proteins further inhibited the maintenance of DNA methylation, confirming the results of the Hi-TransMet (Figures 2-4) and in vitro methylation assays (figure 6b).

Combined, these results strengthened our conclusion that SOX2 alone can demethylate its binding sites. However, our new results indicate that OCT4 also has a capacity, although weaker, to induce demethylation and that there is a synergy between SOX2 and OCT4 activities, resulting in a more efficient DNA demethylation at common binding sites. We modified the text to reflect this fact. We want to thank the reviewer for raising this point as the results made an important addition to the original manuscript.

3. In the fifth section, the author conducted differential expression analysis between ESCs and NPs and found PPFs and SPFs are cell-type specific. However, it seems that this chapter is not that closely related to subsequent chapters.

We did not intend to infer any specific role of SPFs in neuronal differentiation. The main purpose of this section is to further validate the Hi-TransMet assay, before going into the mechanistic part, as the differentiation represents a dynamic state where the expression and activity of the tested PFs are changing. As expected, we observe that the methylation levels at the binding motifs anti-correlate with the expression/activity of the assigned TF (Fig 4 c-d). We now changed the title of this section and changed the text to better communicate the aim of this section.

Minor points:

1. It's hard for readers to understand the method only by watching figure 1. Details or more labels may be appreciated. *We now changed the schematic in figure 1b to make it easier to understand. We also added more explanation both in the first paragraph of the results section and in the*

description of figure 1.

2. In figure 1b, the authors should provide icon of unmethylated CpG and methylated CpG as well as Fig.1a. *Done, accordingly.*
3. The citation of figure 1c needs to be added to the corresponding main text. *Done, accordingly.*
4. Statistical test needs to be added to figure 1d. *Done, accordingly.*
5. In figure 2c,4a,4b and 5c, the gene name should be italic. *Done, accordingly.*
6. In figure 4, the words "Fig.4" are missed. *This has been added.*
7. In figure 4g, the word "g" is missed. *This has been corrected.*
8. In figure 5c and d, the figures are not aligned. *We now aligned the figures.*
9. The X- and Y-axis should be marked in all figures. *All axes are now annotated.*
10. In Supplementary table 1 and 3, in row 1, the content is in the middle of the table, however, in the rest rows, the content is left aligned. *All rows are now left aligned.*
11. In line 124, two different directionalities should be written as two different directions. *The term orientation was now used.*
12. In line 186, "also" should be deleted. *The word "also" was moved behind the PPF names.*
13. In line 268, an extra punctuation mark is used. *The extra punctuation was removed.*
14. In line 643, "WT: CCGTAGTCGA and SC: TCGAGCAGTC", are the bold letters necessary? *This is to emphasize that the CG position was maintained.*
15. In line 653, "Library Data Processing" should be written as "Library data processing". *Done, accordingly.*
16. The logic discontinuous from "SOX2 inhibits DNMT1 activity" to "SOX2 inhibits DNA methylation maintenance during replication". *We replaced the second sentence by "SOX2 inhibits DNMT1-dependent maintenance of DNA methylation during replication".*

Reviewer #2 (Remarks to the Author):

Vanzan et al used a synthetic library approach to identify pioneer factors that lead to changes in the local changes in DNA methylation. They carried out this experiment in ES cells as well as ES-derived NP cells. The authors then conducted genetic and biochemical experiments to probe the mechanism used by these factors to affect DNA methylation, in particular, their relation to TET and DNMT1.

There is an impressive amount of work in this study. The approach is novel, and the results are interesting. However, a few major concerns need to be addressed first:

We are pleased that the reviewer appreciates the novelty and importance of the results, as well as the amount of work that was put in the study.

1. The assumption that the PFs bind to their engineered motifs are concerning. It is known that TFs, and even PFs, only bind to a fraction of their recognition sites in the genome in a way that

we do not fully understand. The CHIP study in Supplementary Figure 6 only tested factors that were tested positive in the assay. Even in these cases, significant binding of some factors (like NRF1) was not detected. The authors should test a few factors that did not show demethylation effect. If it turns out that these factors cannot bind to the engineered motifs, then the conclusion “most PFs do not induce changes in DNA methylation at their binding sites” are not valid.

We thank the reviewer for his insightful comment. We indeed agree that it is not guaranteed that each inserted WT motif will be able to recruit its assigned TF, therefore we consider our approach as explorative instead of exhaustive. In other words, although we identify TFs which binding affect DNA methylation, we cannot completely rule out that the TFs that do not have an effect, are unable to bind to unmethylated WT motifs. We added a sentence in the discussion to reflect this fact. We also changed the text in several places for the same purpose. Moreover, we provided additional in silico and experimental evidence to address this point (see response to point 1 of reviewer 1).

2. Because the factor binding and motifs may not have one to one relation, knockdown experiment should be performed on selected factors. At least for Sox2, evidence should be presented that methylation level goes up after Sox2 knockdown.

Following the excellent suggestion of the reviewer, we knocked down OCT4, SOX2 or both in the RMCE cell line containing the methylated OCT4SOX2 motif (+Sssl condition), followed by Hi-TransMet (see response to point 2 of reviewer 1).

3. There has been reports that the DNA methylation level is correlated with nucleosome occupancy. As pointed out in the manuscript, PF can lead to decreased nucleosome occupancy near its binding site. Could this effect a side effect of nucleosome occupancy change? If this question is beyond the scope of the current manuscript, at least some discussion should be made.

To investigate whether PF binding generates chromatin accessibility by reducing nucleosome occupancy, we performed ATAC-Seq and ATAC-qPCR on RMCE cell lines containing CTCF or OCT4SOX2 motifs (WT/Sc/-Sssl/+Sssl). Results showed that although CTCF, SOX2, and OCT4 can bind their WT motifs (+/- Sssl, figures 1d, 5c and supplementary figure 12), they failed to significantly increase the accessibility of the chromatin at their binding sites. These results indicate that a single PF may not be sufficient to induce changes in chromatin accessibility and would need collaborative binding of either additional molecules of the same PFs or other TFs to specific nearby sequences, as was reported in previous studies (Chronis et al., 2017, Mayran et al., 2019, Sönmezer et al., 2020).

4. The effect presented are on the synthetic DNA. Does this effect apply to the native genome? Some bioinformatics analysis should be done to show if DNA methylation is protected at the binding sites of the identified factors.

A similar analysis as what the reviewer suggests was performed by Stadler et al., 2011.

In addition, following the reviewer's suggestion, we now performed an analysis in which we overlapped methylation data in ES cells to aggregate PF mouse cistromes, the maps of genomic regions that are routinely identified as PF binding sites (Vorontsov et al., 2015). Results of this analysis and the published results in Stadler et al both indicate that, as expected, most PFs binding sites have low methylation. It is important to note that these results are correlative and do not distinguish whether the loss of methylation happened after or before TF binding.

5. The effect of Tet1 and DNMT1 are done with different method – genetic vs biochemical. Rationale should be provided why these two enzymes are treated differently.

While DNMT1 KO and DNMT TKO mESCs exist and are viable (Liao et al., 2015, Tsumura et al., 2006), they would not be a useful model to study passive DNA demethylation, because lack of DNMT1 will result in a global loss of DNA methylation, not limited at specific SPFs binding sites. Hence, we resorted to a biochemical assay (in vitro replication assay).

6. For the biochemical reaction, how are the concentrations of the factor determined? Is there some estimate of the Kd?

For commercially obtained proteins, we used the concentration provided. For proteins prepared in house, the protein concentration was measured by Bradford assay, and a protein gel used to verify the concentration and correct for the purity of the prep. We did not measure Kds. An estimate of the Kd of Sox2 for one of the 4 preps used based on EMSA is ~ 2nM for a specific binding site and ~30X higher for a non-specific site. We also measured the fraction of active proteins in one of our preparations and found it to be 0.1-0.2. Thus, we estimate the Kd to be <1nM. Our EMSA probe was used at 1nM in the Kd assay, so this should be regarded as an upper limit. This has two implications for the replication-coupled DNA methylation assays: 1) the stated concentrations are likely 5-10X higher than the active concentrations; 2) it is likely that in all cases that both the protein and the template are well above the Kd in the DNA replication assays. Because protein is always in excess of DNA binding sites, most of the templates are likely to be bound. This is evident in the EMSA carried out before the start of DNA replication (supplementary figure 10e). Because all of the proteins tested were used at similar high concentrations, and all bind the DNA replication template, we believe the conclusion that Sox2 blocks DNA replication-coupled DNA methylation is valid.

Minor points:

There are many abbreviations in the manuscript that makes it hard to read. Terms like “triple knockout strain” or “hypomethylated regions”, do they have to be shown in abbreviations? *We tried to reduce the use of abbreviations to the minimum in the revised version, while respecting the word limitation.*

REVIEWERS' COMMENTS

Reviewer #1 (Remarks to the Author):

It is pleasant to receive this revision. The revised manuscript is much improved, especially the manuscript writing. I commend the authors on their efforts to reply to the suggestions about the reliability of the proposed method, the specification of roles of OCT4 and SOX2, and other additional data requested.

Here are some additional suggestions for the authors:

1. The meaning of "%input" should be explained in the figure legend for the results of all ChIP assay. Dose "input% < 1" mean that there was no enrichment of that TF on the selected motif?
2. It would be helpful for readers to quickly understand the figures without reading the legends if the authors provide more labels on the figures. For example, adding "CpG positions" in the bottom of Fig1c, Fig2a, Fig2b, Fig3a, Fig3b would make these figures easier to understand. Besides, for F1d, it would be better if the title of the y-axis is "Enrichment of CTCF (input%)".
3. Fig4c, d, better to label the conditions "-SssI/ +SssI", Fig4g can be adjusted better
4. The titles of Supplementary Figure 8 and 9 are missed.

Reviewer #2 (Remarks to the Author):

The authors did a nice job addressing my concerns. I appreciate the effort of the additional experiments -- the knockdown and the ATACseq, and I found the ATACseq data quite interesting. I recommend this manuscript for publication.

REVIEWERS' COMMENTS

Reviewer #1 (Remarks to the Author):

It is pleasant to receive this revision. The revised manuscript is much improved, especially the manuscript writing. I commend the authors on their efforts to reply to the suggestions about the reliability of the proposed method, the specification of roles of OCT4 and SOX2, and other additional data requested.

We thank the reviewer for taking the time to read and comment this revised manuscript. Below are point-by-point answers to the reviewer's comments.

Here are some additional suggestions for the authors:

1. The meaning of “%input” should be explained in the figure legend for the results of all ChIP assay. Dose “input% < 1” mean that there was no enrichment of that TF on the selected motif?

In all ChIP-qPCR experiments, binding of selected TFs to their motif in the FR1 fragment was measured as percentage of input, that is, enrichment of IP signal at a selected locus (FR1), normalized over input signal at the same locus. We estimated TFs enrichment at their WT motifs relative to their enrichment at their corresponding Sc motif. This was significant even in cases when %input<1 (CREB, SOX2). Moreover, the increase in binding at the CREB WT motif (but not at the Sc one) in NPs compared to ESCs was significant even if %input<1.

2. It would be helpful for readers to quickly understand the figures without reading the legends if the authors provide more labels on the figures. For example, adding “CpG positions” in the bottom of Fig1c, Fig2a, Fig2b, Fig3a, Fig3b would make these figures easier to understand. Besides, for F1d, it would be better if the title of the y-axis is “Enrichment of CTCF (input%)”.

We thank the reviewer for the helpful suggestions. The label “CpG positions” was added to all relevant panels and the title of the y-axis has been changed in Fig.1d and Sup.Fig.12a-h.

3. Fig4c, d, better to label the conditions “-Sssl/ +Sssl”, Fig4g can be adjusted better

“-Sssl/+Sssl” labels were added to Fig.4c-f and Fig.4g was adjusted.

4. The titles of Supplementary Figure 8 and 9 are missed.

Titles have been added to legends of Supplementary Figures 8 and 9.

Reviewer #2 (Remarks to the Author):

The authors did a nice job addressing my concerns. I appreciate the effort of the additional experiments -- the knockdown and the ATACseq, and I found the ATACseq data quite interesting. I recommend this manuscript for publication.

We thank the reviewer for taking the time to read and revise this manuscript, and for the helpful suggestions that contributed to increase its quality.